# Interpretable Oracle Bone Script Decipherment through Radical and Pictographic Analysis with LVLMs

## Abstract

As the oldest mature writing system, Oracle Bone Script (OBS) has long posed significant challenges for archaeological decipherment due to its rarity, abstractness, and pictographic diversity. Recently, deep learning-based methods have made exciting progress on the OBS decipherment task. However, they often ignore the intricate connections between the glyphs and meanings of OBS, resulting in limited generalization and interpretability. To this end, we propose an OBS decipherment method based on Large Vision-Language Models, which attempts to bridge the gap between glyphs and meanings and to interpret the deciphering process. Specifically, we propose a progressive training strategy that guides the model from radical analysis to pictographic analysis and then to mutual analysis, enabling it to comprehend the rich semantic information embedded within OBS glyphs. These analysis contents are used to obtain decipherment results (i.e., the corresponding modern Chinese characters), retrieved from a dictionary via our proposed Radical-Pictographic Dual Matching mechanism, thereby allowing the decipherment process to be interpretable. To facilitate model training, we also propose a Pictographic Decipherment OBS Dataset, which comprises 3,173 OBS classes and 47,157 Chinese characters from different dynasties, which is a well-organized dataset containing detailed glyph analysis. Experiments on public benchmarks demonstrate that our method achieves competitive OBS decipherment capabilities and interpretability. Additionally, the interpretability enables our method to provide possible applicable reference content for undeciphered OBS, and thus has potential applications in historical research. The dataset and code repository will be released in camera-ready.

## 1 Introduction

Oracle Bone Script (OBS) is the earliest known mature writing system, inscribed on turtle plastrons and animal bones, and often resemble the shapes of real-world objects. Due to their significant importance in archaeology and history, deep learning-based methods for deciphering OBS have garnered considerable attention recently. These methods aim to predict the corresponding modern Chinese characters for OBS, particularly the OBS not encountered during training.

However, the OBS decipherment task remains a formidable challenge due to the rarity, abstraction, and diversity of its glyphs and the lack of complete contextual information. Over 4,500 unique oracle bone characters have been discovered, yet only about one-third have been successfully deciphered. Early classification model-based approaches Guo et al. (2022); Luo et al. (2023); Zheng et al. (2024); Gan et al. (2023); Lin et al. (2022); Jiang et al. (2023) primarily relied on CNN or Transformer-based visual backbones to perform the classification task for predicting the corresponding modern Chinese characters. Despite their effectiveness, these methods struggle to handle unseen OBS, *i.e.*, zero-shot settings, severely limiting their applicability and failing to achieve true *'decipherment'*. In recent years, composition-based methods Shi et al. (2025); Wang et al. (2024b); Hu et al. (2024; 2025) have been proposed to handle the OBS decipherment task by attempting to decompose OBS into sub-components. These methods predict the corresponding modern format for each component and reassemble them to predict the final corresponding modern Chinese, endowing these approaches with better zero-shot capability and interpretability. Diffusion-based methods Guan et al. (2024b);

Figure 1: Qualitative summary of three existing paradigms and our paradigm for OBS decipherment. "Poor / Medium / Good" reflects trends in Table 1: Accuracy and Zero-shot correspond to the validation and zero-shot metrics. Interpretability is defined by the level of intermediate analysis: no interpretable cues (Poor), coarse structural hints (Medium), and fine-grained reasoning (Good).

Li et al. (2023) for OBS decipherment have been proposed recently, achieving significant advances in both accuracy and zero-shot capability through conditional control and sampling strategies. Despite the significant progress made, all the above methods overlook the rich associations between pictographic forms and semantic information inherent to OBS, resulting in suboptimal decipherment accuracy and insufficient interpretability. Multiple studies on OBS Qiao et al. (2024); Li et al. (2025a) have demonstrated that the semantic information conveyed by radical glyphs often determines the fundamental meaning of a character, and that the pictographs are also highly correlated with semantic contents. Therefore, radical and pictographic information may be highly beneficial to decipher OBS and interpret the decipherment process, which is overlooked by existing methods. To this end, we propose to bridge the glyphs and meanings of OBS using the powerful cross-modal reasoning ability of Large Vision-Language Models (LVLMs) and tailor a progressive training strategy. We first train the model to perform radical recognition and analyze the semantic information embedded in radicals to understand the fundamental meaning of characters. Then, we train the model to perform pictographic analysis for the whole character to grasp the character-level semantic meanings. Finally, we utilize the mutual analysis so that the two levels of analysis complement each other. In addition, we propose a novel Radical-pictographic Dual Matching mechanism, which uses the analysis content to find suitable candidate characters in a dictionary and brings better zero-shot performance. This analysis-to-match process endows our model with the ability to cope with unseen OBS better and explain the logical analysis chains, enhancing the interpretability and generalization of our method.

Although LVLMs have achieved excellent performance on many general tasks, applying them directly to the decipher task and the aforementioned radical and pictographic analysis task is still difficult due to a lack of domain-specific knowledge of OBS. To address this, we introduce a pictographic analysis dataset named PD-OBS. The PD-OBS dataset contains 3,173 Chinese characters annotated with OBS images and detailed radical and pictographic analysis text. Some additional Chinese characters were also collected, bringing the total number of characters to 47,157, and 10,968 characters were annotated with ancient formats. These additional characters are also labeled with analytical text to construct a comprehensive dictionary.

Experiments demonstrate that our method achieves more accurate decipherment with excellent zero-shot capability and decipherment interpretation. The main contributions of this work are as follows:

- We propose an LVLM-based decipherment framework to bridge the gap between glyphs and meanings in OBS, integrating radical and pictographic analyses for OBS decipherment and explicating the decipherment process.

- We designed a progressive training to gradually guide the model in building relationships between glyphs and meanings through radical, pictographic, and mutual analysis. Based on these analyses, we designed a Radical-Pictographic Dual Matching mechanism to replace directly predicting results, achieving better performance, especially for unknown OBS.

- We propose the PD-OBS dataset containing multiple OBS images and related ancient and modern characters, annotated with detailed radical and pictographic analysis from authoritative classical dictionaries, providing a well-structured benchmark for OBS research.

- Our method achieves competitive performance on both oracle bone recognition and decipherment tasks, significantly outperforms existing approaches in zero-shot Top-10 accuracy, and additionally offers fine-grained interpretability. This well-balanced combination of accuracy, zero-shot generalization, and interpretability makes our approach highly promising for applications in related fields.

## 2 RELATED WORK

### 2.1 ORACLE BONE SCRIPT DATASETS

With the continuous excavation of OBS and the steady expansion of digitized resources, an increasing number of high-quality datasets Li et al. (2020); Hu et al. (2025); Han et al. (2020b); Huang et al. (2019); Yue et al. (2022b); Chen et al. (2025a); Li et al. (2026; 2025b); Diao et al. (2025) have been curated and released as open-access resources. Since the introduction of the first publicly available OBS dataset Oracle-20K Guo et al. (2016), the volume and quality of data have improved significantly. In particular, the release of two large datasets, HUST-OBC Wang et al. (2024a) and EVOBC Guan et al. (2024a), has dramatically expanded the pool of available data. These datasets contain over 70,000 oracle bone character samples covering over 3,000 different Chinese character categories.

Currently, HUST-OBC Wang et al. (2024a) and EVOBC Guan et al. (2024a) are the most widely adopted benchmark datasets for OBS research. The HUST-OBC dataset, derived from books, websites, and the collation of previous datasets, collects 77,064 sample scanned or handwritten images of a total of 1,588 deciphered character classes, as well as 62,989 scanned images of undeciphered samples. The EVOBC dataset contains 229,170 images collected from authoritative literature and websites, containing 13,714 different character classes. These images cover six historical stages of ancient scripts: OBS, Chinese Bronze Inscriptions, Seal Script, Spring and Autumn Period Script, Warring States Period Script, and Clerical Script.

### 2.2 ORACLE BONE SCRIPT DECIPHERMENT

Recently, classification model-based approaches Meng (2017); Zhou et al. (1995); Zheng et al. (2024); Lin et al. (2022); Jiang et al. (2023); Fujikawa et al. (2021); Dosovitskiy et al. (2021); Li et al. (2026; 2025b); Diao et al. (2025) for OBS decipherment have emerged, some of which have demonstrated performance comparable to or even surpassing that of human archaeologists in the closed-set setting. Enhanced Inception-V3 Guo et al. (2022) employs convolutional attention modules instead of standard convolutional layers to improve decipherment performance based on a CNN backbone. Building on Transformer architectures, the Pyramid Graph Transformer Gan et al. (2023) integrates a pyramid-structured Vision Transformer (ViT) with skeleton graph representations, attaining state-of-the-art results in closed-set OBS decipherment.

However, the inability to handle unseen OBS class limits the potential application in addressing undeciphered OBS of classification model-based approaches. In recent years, many methods have attempted to decipher OBS classes that are absent from the training set, which can reflect the model's generalization capability and its potential value for undeciphered OBS. Wang *et al.* Wang et al. (2024b) attempted to decompose the structural components of OBS using segmentation models, followed by clustering methods to align these components with the radicals of modern Chinese characters. Although this method facilitates interpretability and archaeological validation, it fails to account for the significant differences in glyph structure between OBS and modern Chinese, resulting in limited decipherment accuracy. The diffusion-based OBSD Guan et al. (2024b) establishes an efficient transforming between ancient characters and modern Chinese characters by combining local structure sampling with style adaptation. The method uses ancient characters as conditional inputs to guide modern Chinese character generation, achieving remarkable accuracy. The unpredictable output of OBSD enables its predictions to transcend dictionary constraints, yet it still suffers from instability and a lack of interpretability. Oraclesage Jiang et al. (2024) is the first to employ LVLM for the description and analysis of OBS. However, its accuracy remains suboptimal, primar-

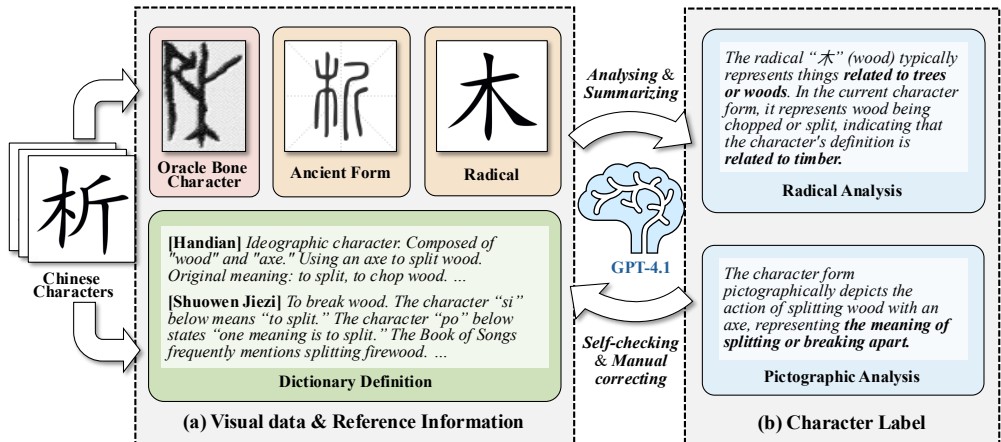

Figure 2: The demonstration of our data engine.

ily due to the insufficient exploitation of glyph features and existing dictionary resources. Therefore, we propose to bridge the glyphs and meaning based on LVLMs and propose a large pictographic decipherment dataset to adapt LVLMs to OBS, enhancing the accuracy, generalization, and interpretability of OBS decipherment.

## 3 PICTOGRAPHIC DECIPHERMENT OBS DATASET

**Dataset Collection.** As mentioned above, existing LVLMs still face a significant challenge when applied to the OBS decipherment task despite their excellent performance on multiple general tasks. To alleviate this challenge, we introduce the Pictographic Decipherment OBS (PD-OBS) dataset to train LVLMs with the capability for analyzing radical and pictographic, which is crucial for the OBS decipherment task. The PD-OBS dataset comprises a total of 47,157 Chinese characters. Among these, 3,173 characters are associated with OBS images collected from the public HUST-OBC and EVOBC datasets; 10,968 characters are provided with ancient Clerical Script images from glyph repositories; and all characters are accompanied by modern regular script images from Han Dian. In addition to image data, each character is annotated with detailed radical analysis and pictographic analysis using text, which are closely related to the semantic meaning of the character. It is worth noting that the original annotations were based on Chinese due to the inclusion of numerous ancient and modern Chinese characters. However, we have translated them into English for presentation purposes throughout the main text and appendix.

**Dataset Annotation.** The annotation process is conducted in three stages, as illustrated in Figure 2. First, we retrieve radical labels, definitions, and explanations for each character from Shuowen Jiezi (an ancient Chinese dictionary) and the authoritative dictionary Han Dian. Second, we associate the acquired root labels and their explanations with each character's modern, ancient script, and OBS image. We further utilize GPT-4.1 OpenAI (2024) to enrich the radical labels based on the referenced glyph images and to summarize the analysis contents. Finally, both automated self-checking with GPT-4.1 and manual review are performed to correct non-standard labels or deviate from the actual character meanings.

**Dataset Usage.** The dataset plays a foundational role and is utilized in two key stages of our method: We construct multi-modal, multi-turn dialogue training samples by pairing OBS images with corresponding modern character labels, enhancing the LVLM's basic capacity to understand OBS glyphs. We group all characters by their radical tags and use a BERT model Devlin et al. (2019) to encode the character label text into feature vectors, forming a Chinese character–pictograph analysis dictionary $\mathcal{D}$ that serves as a reference for matching and verifying decipherment outputs. More details of the dataset are presented in the **supplementary materials**.

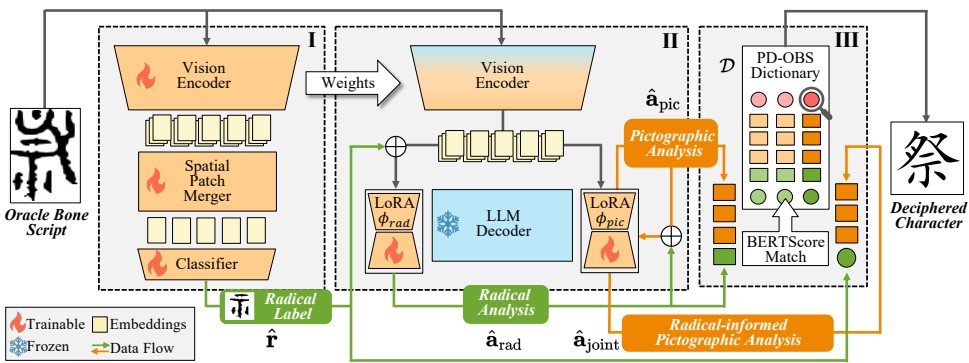

Figure 3: The framework of the proposed method. **I** is used to indicate the Radical Recognition stage, and **II** is used to indicate the Radical-Pictographic Mutual Analysis stage, while **III** is used to indicate Radical-Pictographic Dual Matching.

# 4 METHOD

## 4.1 FRAMEWORK

Our framework is built upon Qwen2.5-VL-7B Bai et al. (2025), sharing the same vision encoder and LLM module. As illustrated in Figure 3, we introduce a spatial patch merger as the visual adapter and a classifier to predict the radical label. We also propose a radical LoRA module $\phi_{\text{rad}}$ and a pictographic LoRA module $\phi_{\text{pic}}$ Hu et al. (2021) to analyze the corresponding information. Furthermore, we design a progressive training—starting from radical recognition (Sec. 4.2), followed by radical and pictographic analysis (Sec. 4.3), and culminating in mutual analysis—to gradually lead the model to bridge the gap between glyphs and meanings of OBS. Finally, we propose a novel radical-pictographic dual matching mechanism (Sec. 4.4) to select the appropriate modern Chinese characters from the database as the decipherment result. More training details and analysis contents can be found in the **Appendix**.

## 4.2 RADICAL RECOGNITION

In this stage, we aim to adapt the vision encoder to the unique visual style of OBS and to predict radical labels that will serve as critical cues for downstream reasoning. For this purpose, we designed a spatial patch merger as a visual adapter, which downsamples the visual embeddings to obtain a representative feature vector for classification tasks. In addition, we design a triplet loss Schroff et al. (2015) based on the Euclidean distance to explicitly improve the differentiation of feature vectors with different radicals.

Specifically, we implement a sampling strategy to ensure that each batch contains at least two samples for each radical class. During training, for each sample in the batch, we designate its feature vector $\mathbf{v}_n$ as an anchor sample, then select a positive sample $\mathbf{v}_n^+$ (*i.e.*, a sample with the same radical label) and a negative sample $\mathbf{v}_n^-$ (*i.e.*, a sample with the different radical label). The triplet loss is as follows:

$$\mathcal{L}_{\text{trip}} = \frac{1}{N} \sum_{n=1}^{N} \max\left( \|\mathbf{v}_n - \mathbf{v}_n^+\|_2 - \|\mathbf{v}_n - \mathbf{v}_n^-\|_2 + \alpha, \ 0 \right), \tag{1}$$

where $N$ is the number of triplets in the batch, $\|\cdot\|_2$ denotes the Euclidean ($\ell_2$) norm, and $\alpha$ is the margin hyperparameter.

Regarding the classifier, we use the cross-entropy loss $\mathcal{L}_{\text{ce}}$ to optimize it. Therefore, the whole loss function $\mathcal{L}_{\text{stage1}}$ of this stage can be shown as follows:

$$\mathcal{L}_{\text{stage1}} = \mathcal{L}_{\text{ce}} + \gamma \mathcal{L}_{\text{trip}}, \tag{2}$$

where $\gamma$ is a hyperparameter used to balance the two items.

**Algorithm 1** Radical-Pictographic Dual Matching

**Require:** Dictionary $\mathcal{D}$
$\quad \mathcal{D} = \{(\mathbf{r}_i, \mathbf{a}_{\mathrm{rad},i}, \mathbf{a}_{\mathrm{pic},i}, \mathbf{a}_{\mathrm{joint},i}, y_i)\}_{i=1}^N$
**Require:** Model output $(\hat{\mathbf{r}}, \hat{\mathbf{a}}_{\mathrm{rad}}, \hat{\mathbf{a}}_{\mathrm{pic}}, \hat{\mathbf{a}}_{\mathrm{joint}})$
**Require:** Parameter $k$ (Top$k$)
**Require:** $\mathcal{S}(\cdot, \cdot)$: Semantic similarity between two texts calculated using the BERT-Score.
1: // Filtered Matching
2: $\mathcal{D}_{\mathrm{rad}} \leftarrow \{i \mid \mathbf{r}_i = \hat{\mathbf{r}}\}$
3: $C_1 \leftarrow$ Top-$k$ indices in $\mathcal{D}_{\mathrm{rad}}$ by $\mathcal{S}(\mathbf{a}_{\mathrm{pic},i}, \hat{\mathbf{a}}_{\mathrm{pic}})$
4: // Joint Matching
5: $C_2 \leftarrow$ Top-$k$ indices in $\{1, \ldots, N\}$ by $\mathcal{S}((\mathbf{a}_{\mathrm{rad},i} \oplus \mathbf{a}_{\mathrm{joint},i}), (\hat{\mathbf{a}}_{\mathrm{rad}} \oplus \hat{\mathbf{a}}_{\mathrm{joint}}))$
6: // Dual Matching
7: $C \leftarrow C_1 \cup C_2$
8: $R \leftarrow$ Top-$k$ in $C$ by their similarity scores
9: **return** $\{y_i \mid i \in R\}$

Figure 4: The workflow of radical-pictographic mutual analysis.

## 4.3 RADICAL-PICTOGRAPHIC MUTUAL ANALYSIS

To bridge glyphs and meaning in OBS, we design a progressive glyph analysis process to facilitate the decipherment task. Specifically, we introduce a progressive training procedure, beginning with radical analysis, where the radical is predicted in the radical recognition stage. In both OBS and ancient Chinese characters, radicals often determine the basic semantics of characters, as illustrated by the Q1&A1 in Figure 4. Therefore, we train the model's radical analysis capability with a large number of radical–analysis Q&A pairs constructed from the PD-OBS dataset. Next, we guide the model to perform a pictographic analysis for the entire character to analyze the meaning embedded in the full character glyph, as shown by the Q2&A2 in Figure 4.

In our observations, direct pictographic analysis of LVLM may yield erroneous results, likely due to the lack of prior knowledge or the overall abstract nature of the script, as shown in the A2 of Figure 4. Therefore, we design a mutual analysis as the final step, which informs the pictographic analysis with insights from radical analysis, resulting in more accurate character meanings. Specifically, we employ radical analysis $\hat{\mathbf{a}}_{\mathrm{rad}}$ and pictographic analysis $\hat{\mathbf{a}}_{\mathrm{pic}}$ as contextual information, prompting LVLM to re-examine pictographic analysis and generate the radical-informed pictographic analysis $\hat{\mathbf{a}}_{\mathrm{joint}}$ as illustrated by Q3&A3 in Figure 4. This enables the model to explicitly consider the basis of radical-implicated information, thereby mitigating the difficulty of directly analyzing the entire character.

During training, we initialize the visual encoder with the weights from the previous stage, freezing the shallow layers to retain low-level features while fine-tuning the deeper layers for high-level semantic adaptation. In addition, we introduce a radical LoRA module $\phi_{\mathrm{rad}}$ and a pictographic LoRA module $\phi_{\mathrm{pic}}$ Hu et al. (2021), the former for radical analysis while the latter for pictographic and mutual analysis. The training data consists of Q&A pairs from the PD-OBS dataset, as illustrated in Figure 4, and the loss function employed is the cross-entropy loss commonly used in LVLM training.

## 4.4 RADICAL-PICTOGRAPHIC DUAL MATCHING

Based on the above two stages, we obtain four intermediate results for each test character: predicted radical label $\hat{\mathbf{r}}$, radical analysis $\hat{\mathbf{a}}_{\mathrm{rad}}$, pictographic analysis $\hat{\mathbf{a}}_{\mathrm{pic}}$, and radical-informed pictographic analysis $\hat{\mathbf{a}}_{\mathrm{joint}}$. We propose a dictionary-based dual matching mechanism for decipherment. Given the candidate dictionary $\mathcal{D} = \{(\mathbf{r}_i, \mathbf{a}_{\mathrm{rad},i}, \mathbf{a}_{\mathrm{pic},i}, \mathbf{a}_{\mathrm{joint},i}, y_i)\}_{i=1}^N$ from the PD-OBS dataset, in which $y_i$ symbolizes modern Chinese characters, the mechanism works as follows:

First, we filter candidates by the predicted radical label $\hat{\mathbf{r}}$, then select the Top-$k$ entries by the semantic similarity $\mathcal{S}(\mathbf{a}_{\mathrm{pic},i}, \hat{\mathbf{a}}_{\mathrm{pic}})$ calculated by BERT-Score Zhang et al. (2020) between the pictographic analyses. Second, we concatenate the radical analysis and radical-informed pictographic analysis, and select another Top-$k$ entries by similarity $\mathcal{S}((\mathbf{a}_{\mathrm{rad},i} \oplus \mathbf{a}_{\mathrm{joint},i}), (\hat{\mathbf{a}}_{\mathrm{rad}} \oplus \hat{\mathbf{a}}_{\mathrm{joint}}))$. Finally, we

Table 1: Each cell reports Top-1 / Top-10 accuracy (in %). The best and second-best results are respectively marked in bold and underlined. *Improvement* represents the performance gains achieved by our method compared to the existing best method.

| Method | Validation | | Zero-shot | |
|---|---|---|---|---|
| | HUST-OBC | EVOBC | HUST-OBC | EVOBC |
| *classification model-based* | | | | |
| InceptionV3 Guo et al. (2022) | 74.4 / 76.9 | 62.4 / 64.5 | - / - | - / - |
| ViT Dosovitskiy et al. (2021) | 79.2 / 81.7 | 72.7 / 74.2 | - / - | - / - |
| PyGT Gan et al. (2023) | **84.3** / 87.6 | **78.1** / 81.2 | - / - | - / - |
| *Composition-based* | | | | |
| PPP Wang et al. (2024b) | 76.8 / - | 72.4 / - | 13.6 / - | 19.1 / - |
| *Commercial LVLM* | | | | |
| GPT-4.1 OpenAI (2024) | 6.0 / 10.4 | 4.5 / 8.4 | 5.3 / 7.3 | 4.3 / 9.2 |
| Qwen-VL-Max Bai et al. (2025) | 4.8 / 6.6 | 4.1 / 5.6 | 2.0 / 2.5 | 4.0 / 6.2 |
| Gemini-2.5-Pro Gemini Team, Google (2025) | 6.3 / 13.9 | 5.1 / 10.4 | 5.0 / 8.6 | 6.2 / 15.0 |
| GPT-5 OpenAI (2025) | 7.2 / 16.1 | 5.3 / 12.5 | 6.4 / 9.8 | 6.0 / 14.3 |
| *Diffusion-based* | | | | |
| OBSD Guan et al. (2024b) | 66.8 / 72.9 | 71.2 / 77.9 | **18.3** / 27.5 | 30.4 / 50.5 |
| BBDM Li et al. (2023) | 55.8 / 59.5 | 60.3 / 62.1 | 8.0 / 14.1 | 19.5 / 29.5 |
| **Ours** | 80.6 / **87.8** | 76.3 / **81.7** | 16.8 / **53.7** | **33.3** / **64.1** |
| *Improvement* | -3.7 / +0.2 | -1.8 / +0.5 | -1.5 / +26.2 | +2.9 / +13.6 |

merge and re-rank these candidate sets to obtain the Top-$k$ modern Chinese characters as decipherment results. All steps and notations are detailed in Algorithm 1.

Notably, we employ the matching mechanism instead of directly outputting decipherment results, which helps mitigate the limited generalization of the model for zero-shot settings and undeciphered OBS caused by the absence of such OBS in the training data.

## 5 EXPERIMENTS

We analyze the primary experimental results in the main text. Due to space constraints, more visualizations, comparative results, ablation studies, and other findings are presented in the **Appendix**.

### 5.1 IMPLEMENTATION DETAILS

All training and evaluation experiments are conducted on 8 NVIDIA RTX 4090 GPUs. We initialize our model with the pretrained weights of Qwen2.5-VL-7B. During the radical recognition stage, we set the learning rate to 5e-4, batch size $N$ to 8, and train for 5 epochs. The hyperparameters $\gamma$ and $\alpha$ in the loss functions are set to 5 and 0.25, respectively. For the radical-pictographic mutual analysis stage, we use a learning rate of 5e-5, batch size of 4, and train for 4,000 steps. AdamW Loshchilov & Hutter (2019) is used as the optimizer. The radical LoRA $\phi_{\mathrm{rad}}$ and pictographic LoRA $\phi_{\mathrm{pic}}$ are configured with a dropout rate of 0.05 and 0.25, respectively, and both use rank and alpha values of 32.

### 5.2 DATASETS AND EVALUATION METRICS

We perform experiments on the commonly used HUST-OBC Wang et al. (2024a) and EVOBC Guan et al. (2024a) datasets. To avoid the complexity of requiring expert verification for undeciphered OBS, we followed OBSD Guan et al. (2024b) by evaluating on previously deciphered inscriptions and excluded test categories from training to ensure their genuine novelty. We select 200 character classes from each dataset as the unknown class (*i.e.* zero-shot test sets). The remaining data are ran-

domly split into training and validation sets in a 9:1 ratio to assess the OBS recognition capabilities on known classes.

We use Top-k accuracy as an evaluation metric, as in previous work Guan et al. (2024b); Gan et al. (2023); Jiang et al. (2024); Chen et al. (2025b), which is usually used in diverse classification tasks Dosovitskiy et al. (2021); Luo et al. (2023); Lin et al. (2022). To evaluate the interpretability of the model, we quantify the consistency between the analyzes content generated and the annotations. Inspired by evaluation practices in image captioning Galliena et al. (2025); Wang et al. (2022) and abstractive summarization Liu et al. (2022), we adopt ROUGE-L Lin (2004), METEOR Banerjee & Lavie (2005), and BERT-Score Zhang et al. (2020) to provide a complementary and holistic assessment of the generated analyzes.

## 5.3 MAIN RESULTS

**Decipherment Result.** To evaluate the effectiveness of our method on the OBS decipherment task, we conduct comprehensive comparisons as shown in Table 1. It should be clarified that in OBS decipherment research, the primary evaluation focus is the zero-shot setting Guan et al. (2024b); Li et al. (2025a), which reflects a model's ability to handle previously unseen characters—the core difficulty in real archaeological scenarios. By contrast, validation performance primarily indicates recognition capabilities for seen characters rather than decipherment ability, serving as a secondary auxiliary metric. We adopt InceptionV3 Guo et al. (2022), ViT Dosovitskiy et al. (2021), and PyGT Gan et al. (2023) as classification model-based baselines, and OBSD Guan et al. (2024b) and BBDM Li et al. (2023) as diffusion-based methods. In addition, we include strong commercial LVLMs, GPT-4.1 OpenAI (2024), Qwen-VL-Max Bai et al. (2025), Gemini-2.5-Pro Gemini Team, Google (2025), and GPT-5 OpenAI (2025) for comparison. However, commercial LVLMs perform poorly in both settings, with Top-1 accuracy consistently below 8%, highlighting their limited capability to understand OBS. On the validation set, although our method yields slightly lower Top-1 accuracy than the best classification model-based baseline (e.g., PyGT), it achieves the highest Top-10 accuracy, demonstrating superior capability in generating high-quality candidates. In the more important and challenging zero-shot scenario, our method exhibits notably strong performance: It remains competitive in Top-1 accuracy with the SOTA method OBSD and significantly outperforms all methods in Top-10 accuracy, surpassing the second-best method by 26.2% on HUST-OBC and 13.6% on EVOBC. These results confirm our method's strong generalization and transferability to unseen OBS, highlighting its potential value in assisting the recognition of undeciphered OBS in archaeological research.

Table 2: Interpretability performance comparison between different methods based on the ROUGE-L / METEOR / BERT-Score.

| Method | HUST-OBC | | EVOBC | |
|---|---|---|---|---|
| | *Validation* | *Zero-shot* | *Validation* | *Zero-shot* |
| Qwen2.5-VL-7B | 0.355 / 0.358 / 0.694 | 0.309 / 0.301 / 0.651 | 0.341 / 0.350 / 0.683 | 0.337 / 0.348 / 0.679 |
| Qwen-VL-Max | 0.391 / 0.402 / 0.705 | 0.335 / 0.334 / 0.656 | 0.378 / 0.383 / 0.698 | 0.359 / 0.355 / 0.682 |
| GPT-4.1 | 0.465 / 0.477 / 0.737 | 0.407 / 0.412 / 0.675 | 0.429 / 0.434 / 0.714 | 0.413 / 0.419 / 0.709 |
| Gemini-2.5-Pro | 0.486 / 0.501 / 0.745 | 0.421 / 0.419 / 0.712 | 0.436 / 0.447 / 0.713 | 0.529 / 0.538 / 0.749 |
| GPT-5 | 0.572 / 0.575 / 0.783 | 0.470 / 0.468 / 0.725 | 0.520 / 0.521 / 0.755 | 0.498 / 0.501 / 0.740 |
| Ours | **0.914 / 0.907 / 0.946** | **0.550 / 0.525 / 0.794** | **0.887 / 0.884 / 0.937** | **0.576 / 0.586 / 0.849** |

**Interpretability Performance.** To quantitatively evaluate the interpretability of our method, we employ Rouge-L Lin (2004), METEOR Banerjee & Lavie (2005), and BERT-Score Zhang et al. (2020) to measure the similarity between the analysis text of Top-1 outputs and the ground truth text from the dictionary $\mathcal{D}$. We evaluate LVLMs, including Qwen2.5-VL-7B, Qwen-VL-Max, GPT-4.1, Gemini-2.5-Pro, and GPT-5, and compare their average performance with our method. As shown in Table 2, our method significantly outperforms the powerful commercial LVLM GPT-5 and Gemini-2.5-Pro across three metrics. This result indicates that the analysis generated by our method is more reliable and informative.

| Data Type | OBS Image | GT | OBSD Top-3 Results | Our Top-3 Results | Radical Analysis (Ours) | Pictographic Character Form Analysis (Ours) |
|---|---|---|---|---|---|---|
| #Validation | | 吹 | 吹 [吹] [叹] [咬] | [吹] [喝] [唤] | Radical "口" is related to the mouth or openings; in the current character form it symbolizes **an open mouth** | A person opening their mouth and forcefully exhaling air, representing **the action of blowing** |
| | | 逐 | 逐 [逐] [隧] [遄] | [逐] [追] [追] | Radical "辶" is related to walking or running; in the current character form it represents the **footsteps of a person running** | A pig running with another person chasing behind it, representing the meaning of **pursuit and driving away** |
| | | 陟 | 陟 [陟] [徙] [涉] | [陟] [隥] [算] | Radical "阝" indicates hillsides or elevated terrain, in the current character form it represents **steps** | A person walking step by step up stairs, representing the meaning of **climbing or ascending** |
| #Zero-shot | | 春 | 春 [春] [春] [春] | [春] [膚] [晴] | Radical "日" indicates the sun, in the current character form it represents **sunshine** | A scene of plants growing and all things reviving under the radiant sunshine, representing **the arrival of spring** |
| | | 穆 | 穆 [穆] [穆] [穆] | [梨] [禾] [穆] | Radical "禾" is related to crops and agricultural plants; in the current character form it symbolizes **ripe rice grains** | A mature grain plant, representing the meaning of **a bountiful harvest** |
| | | 妖 | 娇 [媖] [妍] [妹] | [妖] [嬹] [嬿] | Radical "女" is related to feminine qualities, in the current character form it represents a working **woman** | A woman working in wheat fields, representing **the virtuous qualities of women being diligent and thrifty in managing households.** |
| | | 司 | 刁 [可] [司] [刁] | [司] [后] [嘗] | Radical "口" is related to the mouth or openings; in the character form it symbolizes **a person speaking** | An ancient official reading a proclamation, expressing the meaning of **management and execution** |
| | | 叟 | 宁 [宥] [宕] [宂] | [寏] [反] [叟] | Radical "又" indicates hands; in the current character form it appears as the **image of a hand grabbing something** | A hand holding a staff, representing the meaning of **an elderly elder** |
| | | 巡 | 子 [子] [予] [于] | [巡] [尤] [巡] | Radical "巛" is related to rivers, **unrelated to the current character's pictographic meaning** | A person walking around, representing **the meaning of patrol or inspection.** |

Figure 5: Visualization of the decipherment results and the interpretable content. Green rectangles and fonts indicate correct results and content, while red rectangles and fonts indicate errors. The leftmost item of the OBSD Top-3 results represents the corresponding modern Chinese character image generated by this method.

## 5.4 ABLATION STUDY

**Ablation on Radical Recognition Stage.** To evaluate the effectiveness of the proposed radical recognition stage, we use the original vision encoder of Qwen2.5-VL-7B Bai et al. (2025) as the baseline, and incorporate our radical recognition module or a LoRA-based recognition method. Our method introduces a spatial patch merger and the loss function $\mathcal{L}_{\text{trip}}$ on top of the baseline vision encoder, resulting in improvements of 0.9% and 1.2% accuracy on the validation and zero-shot settings, respectively. The LoRA-based recognition method merges the recognition stage with the radical analysis process and training with LoRA-based fine-tuning. The results demonstrate that this method leads

Table 3: Radical recognition accuracy (in %) of different model variants on the HUST-OBC dataset with validation (Valid.) and zero-shot (ZS) settings.

| Method | Valid. | ZS |
|---|---|---|
| Vision Encoder of Qwen | 92.7 | 87.1 |
| + Our Radical Recognition | 93.6 | 88.3 |
| + LoRA-based Recognition | 80.1 | 69.8 |

to a significant drop in radical recognition accuracy and introduces substantial errors in radical analysis; thus, we retain radical recognition as an independent stage in our framework.

**Ablation on the Proposed Modules and Strategies.** To validate the effectiveness of our proposed modules and strategies, we take Qwen2.5-VL-7B Bai et al. (2025) as the baseline and incrementally add each component to form our final model. The Top-1 and Top-10 performance under both validation and zero-shot settings are shown in Table 4. The results demonstrate that Pictographic Analysis fine-tuning (+ Pictographic Analysis) enables good decipherment ability on the validation set but still lacks generalization in zero-shot scenarios. With the introduction of Radical-Pictographic Mutual Analysis (+ Rad&Pic Mutual Analysis), the model's accuracy improves on the validation set, but the increase in zero-shot ability is still minimal. When we explicitly guided mutual analysis using radical recognition results based on the previous model variant, the model (+ Radical Recognition) achieved similar performance improvements. The primary reason for scant progress lies in the model's insufficient generalization capability for directly predicting results, which often prevents it from deciphering unseen characters — a common challenge in zero-shot scenarios of similar tasks Yu et al. (2023). To mitigate this, we design the Radical-Pictographic Dual Matching mechanism specifically for OBS to replace direct prediction. The final model with the dual matching mechanism (+ Rad&Pic Dual Matching) significantly improves the model's zero-shot performance. To validate the necessity of dual matching, we employed $C_2$ from Algorithm 1 (+ joint Matching using $C_2$) for

matching, and the experimental results demonstrated that this strategy significantly underperformed compared to the dual matching mechanism.

## 5.5 QUALITATIVE RESULTS

To further demonstrate the performance and interpretability of our method, we visualize the decipherment results of ours and OBSD Guan et al. (2024b) as shown in Figure 5. The results reveal two significant advantages of our method: higher robustness and stronger interpretability. Both methods can produce correct predictions on the validation set, but our model more consistently generates semantically aligned Top-3 candidates by leveraging analysis contents. Our method exhibits apparent robustness in the more challenging zero-shot scenario, whereas OBSD fails on several complex or infrequent charac-

Table 4: Top-1 and Top-10 accuracy (in %) of our model and its variants on HUST-OBC dataset.

| Method | Validation | | Zero-shot | |
|---|---|---|---|---|
| | Top-1 | Top-10 | Top-1 | Top-10 |
| Qwen2.5-VL-7B | 1.4 | 1.4 | 0.2 | 0.2 |
| + Pictographic Analysis | 52.4 | 52.4 | 1.6 | 1.6 |
| + Rad&Pic Mutual Analysis | 60.3 | 61.4 | 5.2 | 5.4 |
| + Radical Recognition | 64.2 | 64.2 | 6.6 | 6.6 |
| + Rad&Pic Dual Matching | 80.6 | 87.8 | 16.8 | 53.7 |
| + joint Matching using $C_2$ | 69.1 | 80.5 | 14.9 | 45.8 |

ters. Moreover, the radical and pictographic analyses provide human-consistent explanations, such as linking 'nü' (woman radical) to feminine qualities or 'ri' (sun radical) to sunshine. These interpretable outputs justify the predictions and are more reliable and suitable for the OBS decipherment.

## 6 CONCLUSION

We propose an interpretable OBS decipherment framework based on LVLMs. The framework bridges glyphs to meaning through three stages: radical analysis, pictographic analysis, and mutual analysis. With the proposed Radical-Pictographic Dual Matching, our model can filter the appropriate deciphering candidate set from a dictionary based on analysis content, replacing the direct output of deciphering results to achieve better zero-shot performance. Moreover, these generated textual analyses serve as interpretable contents, offering possible references for undeciphered OBS characters, thus holding potential for archaeological applications. We construct the PD-OBS dataset annotated with radical and pictographic analysis texts to support training, providing a valuable resource for future research. Experimental results demonstrate the excellent performance of our method in decipherment accuracy, generalization, and interpretability.

REPRODUCIBILITY STATEMENT

We have made significant efforts to ensure the reproducibility of our work. In the supplementary materials, we provide both the detailed description of dataset composition and construction, as well as partial raw data samples and demonstrations to illustrate the data characteristics. Additional implementation details, including training setups, hyperparameters, and evaluation protocols, are presented in the main text and the appendix. To further facilitate independent verification, the complete source code and processed datasets will be released upon acceptance of the paper.

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

# A  APPENDIX

## A.1  THE USE OF LARGE LANGUAGE MODELS (LLMS)

Large Language Models (LLMs) were only used to assist with grammar checking of the manuscript. In some cases, minor wording suggestions from the LLM were adopted to improve clarity and readability. No part of the research ideation, experimental design, data analysis, or results interpretation relied on LLMs.

## A.2  MORE IMPLEMENTATION DETAILS

In this section, we will provide additional explanations focusing primarily on the Radical Recognition stage (Sec. A.2.1) and training details (Sec. A.2.2). Due to space constraints, these details were not elaborated upon in the main text.

### A.2.1  THE USED RADICAL

In Chinese, the term "radical" encompasses two distinct concepts: radical components (pianpang) and the indexing radical (bushou). Radical components denote the constituent elements of compound Chinese characters, whereas the indexing radical refers to the single representative component used for character retrieval in dictionary compilation. In our work, "radical" specifically refers to the unique indexing radical of each character. Considering the importance of the indexing radical (bushou) in character retrieval due to their typical semantic function, we adopt the indexing radical (bushou) and its analysis as the key conditions for dictionary matching.

### A.2.2  RADICAL RECOGNITION STAGE

**The Spatial Patch Merger.** In this stage, we introduce the Spatial Patch Merger for feature down-sampling. First, the OBS images are uniformly resized to $224 \times 224$, and patch embedding is performed with the default patch size of $14 \times 14$ following Qwen2.5-VL, ensuring that the Vision Encoder outputs 256 tokens representing the visual features of each OBS image. The Spatial Patch Merger rearranges these tokens according to their original spatial positions and then merges tokens using the patch-merger module from Qwen2.5-VL, which consists of an RMSNorm layer Zhang & Sennrich (2019) and an MLP. Each merge operation reduces the number of tokens to one fourth while keeping the channel dimension unchanged. After four merge iterations, a unique global visual token is obtained, which is then transformed into a feature vector through an MLP.

**Batch Sampling.** To facilitate the computation of the triplet loss function $\mathcal{L}_{\text{trip}}$, we designed a specialized batch sampling method. First, OBS samples are grouped by radical label indices. Then, we randomly traverse all radical labels with a step size of half the batch size, sampling two samples per label. We mark sampled samples to prevent duplicate sampling, allowing repeated sampling only when the remaining samples for a radical label are odd in number.

### A.2.3  TRAINING DETAILS

**Radical LoRA.** When training the radical LoRA module $\phi_{\text{rad}}$, our training data Q&A pairs include all OBS with radical analysis, even though some of them are not strongly correlated with their character semantics. This helps train the model to determine the role of radical information in the overall semantic meaning of characters.

**Pictographic LoRA.** When training the pictographic LoRA module $\phi_{\text{pic}}$, we first perform a warm-up phase for one epoch using pictographic analysis Q&A pairs. The model outputs from this warm-up phase, which usually contain incorrect pictographic analyses, are then combined with radical analysis data from PD-OBS and the original Q&A pairs to construct multi-turn dialogue data. In the formal training stage, we mix these multi-turn dialogues with the pictographic analysis Q&A pairs as the training set. This strategy is crucial for equipping the model with the capability for Radical-Pictographic Mutual Analysis.

During the above two training stages, we freeze the parameters of the first 16 layers of the visual encoder and train the last 16 layers of the visual encoder, the patch merger, and the LoRA modules.

### A.2.4 The Evaluation of Commercial LVLM

To evaluate the performance of commercial LVLMs in OBS decipherment task, we randomly select five samples from the training set as test cases following OracleSage Jiang et al. (2024), enabling these models to generate predictions through in-context learning. Each sample comprises three elements: an Oracle bone script image, its pictographic analysis, and the deciphered result. Each prediction consists of the ten most confidently predicted modern Chinese characters alongside their corresponding analysis content. To mitigate the impact of data randomness, we conducted five evaluations and calculated the average of the performances as the final results.

### A.2.5 Annotation of the PD-OBS

There is a substantial temporal gap between modern Chinese characters and oracle bone script. To mitigate this impact, we introduce an intermediate script layer based on ancient characters and use their philological analyses as auxiliary reference. This design provides GPT-4o with richer and more reliable evidence, enabling it to produce more faithful oracle-bone analyses and thereby support the construction of a higher-quality decipherment dataset. We instantiate this intermediate layer with Clerical Script for two reasons: (1) the historical emergence of Clerical Script is broadly contemporaneous with the compilation of the key reference Shuowen Jiezi, so that its glyph forms are naturally aligned with the dictionary analyses we obtain; and (2) compared with earlier scripts (such as bronze inscriptions), Clerical Script has a substantially larger and more diverse surviving corpus, offering broader coverage of character types.

Each Radical and Pictographic Analysis annotation derived from GPT analysis and summarization is re-input into GPT alongside modern character forms, ancient character forms, and comprehensive dictionary analyses (including all reference information available from authoritative ancient sources like Shuowen Jiezi, Kangxi Dictionary, and Han Dictionary). The following self-checks are then performed: a. Whether the annotation description aligns with the character form; b. Whether the annotation description conflicts with dictionary analyses or contains information not covered by them; c. Whether the annotation summarizes the dictionary analysis content. All problematic annotations are regenerated and re-examined. In addition, the final output undergoes manual correction to minimize erroneous analyses.

### A.3 Discussion of Baselines

In this section, we further analyze existing methods, including those listed and unlisted in Table 1.

**Classification-based Methods.** As shown in Table 1, the PyGT Gan et al. (2023) slightly outperforms our approach on the validation set. This can be attributed to PyGT framing the task as a closed-set classification problem, where a fixed number of character classes significantly simplifies the task. Consequently, classification-based methods like PyGT may be considered incapable of achieving true decipherment and have limited potential for deciphering unknown characters; instead, they are better suited for OBS recognition tasks.

**Diffusion-based Methods.** Although our Top-1 zero-shot accuracy on the HUST-OBC dataset Wang et al. (2024a) is slightly lower than that of the OBSD method Guan et al. (2024b), our approach exhibits a distinct advantage in Top-10 accuracy. Notably, in contrast to the instability of results typically observed with diffusion-based methods, our framework offers higher interpretability, enhancing the reliability of the outputs. In addition, our method encompasses key archaeological procedures, including pictographic analysis, radical analysis, and dictionary verification, thereby offering a more professional and interpretable solution.

**Non-open-source Methods.** As an LVLM-based method, OracleSage Jiang et al. (2024) reported zero-shot Top-1 and Top-10 accuracies of 20.2% and 40.9%, respectively, on a dataset composed of HUST-OBC and EVOBC. OracleFusion Li et al. (2025a) proposes structurally constrained semantic typography for oracle bone script. It fuses component/radical layout with semantic cues to produce interpretable typography that supports decipherment. Despite relevance, we have not yet succeeded in fully reproducing these works, as the source code for both methods remains unpublished and certain experimental details are difficult to obtain. This makes it challenging to incorporate them into our comparative methodology.

## A.4 HYPERPARAMETER SENSITIVITY ANALYSIS

To validate the impact of hyperparameters $\alpha$ and $\gamma$, we conducted a sensitivity analysis as shown in the Table 5. We observe that small $\alpha$ makes the triplet constraint insufficiently strong, leading to weaker separation between visually similar radicals, whereas an excessively large $\alpha$ introduces optimization instability and degrades performance. In addition, since $\gamma$ controls the relative contribution of the triplet loss, $\alpha$ and $\gamma$ need to be adjusted jointly to ensure balanced optimization and prevent either loss from dominating. The empirical trends in the table support the choice of $\alpha$ =0.25 and $\gamma$=5 used in the main experiments.

Table 5: Sensitivity of $\alpha$ (rows) and $\gamma$ (columns). Values are Top-1 / Top-10 accuracy (%).

| $\alpha\backslash\gamma$ | 1 | 5 | 10 |
|---|---|---|---|
| 0.1 | 92.0 / 86.5 | 92.2 / 86.8 | 92.7 / 87.1 |
| 0.25 | 93.1 / 87.7 | **93.6 / 88.3** | 91.4 / 87.9 |
| 0.5 | 91.2 / 86.1 | 89.4 / 84.5 | 88.9 / 83.2 |

## A.5 DISCUSSION OF RADICAL-PICTOGRAPHIC DUAL MATCHING

### A.5.1 ABLATION STUDY ON DUAL MATCHING

To further validate the effectiveness of Radical-Pictographic Dual Matching, we compared it with different matching mechanisms, namely Filtered matching using $C_1$ and Joint matching using $C_2$ as defined in Algorithm 1. As shown in Table 6, both mechanisms perform worse than Radical-Pictographic Dual Matching. This is primarily due to potentially inaccurate pictographic analysis in Filtered matching using $C_1$, while Joint matching using $C_2$ may result in more failure cases on samples where radicals are weakly correlated with character meaning.

Table 6: Decipherment accuracy (in %) under different matching mechanisms.

| @Matching Mechanism | HUST-OBC | | EVOBC | |
|---|---|---|---|---|
| | *Valid.* | *ZS* | *Valid.* | *ZS* |
| Filtered Matching using $C_1$ | 66.6 | 8.8 | 64.6 | 10.3 |
| Joint Matching using $C_2$ | 69.1 | 14.9 | 73.1 | 27.6 |
| Rad&Pic Dual Matching | 80.6 | 16.8 | 76.3 | 33.3 |

### A.5.2 TOP-K PARAMETER ANALYSIS

As shown in Table 7, the zero-shot accuracy of our method significantly improves as we incorporate more characters into the candidate set, achieving a remarkably high accuracy at the Top-50. This result demonstrates that the Radical-Pictographic Dual Matching mechanism can effectively select the appropriate character sets. Although expanding the candidate set may increase the workload, it may enhance the probability of successful decipherment by providing human experts with a more comprehensive reference.

Table 7: Decipherment accuracy (in %) under different Top-k settings. The Valid. and ZS indicate validation and zero-shot settings.

Table 8: Decipherment Top-10 accuracy (in %) under different dictionary scales. The Valid. and ZS indicate validation and zero-shot settings.

| @Top-k | HUST-OBC | | EVOBC | |
|---|---|---|---|---|
| | *Valid.* | *ZS* | *Valid.* | *ZS* |
| Top-1 | 80.6 | 16.8 | 76.3 | 33.3 |
| Top-5 | 86.0 | 39.3 | 79.8 | 56.0 |
| Top-10 | 87.8 | 53.7 | 81.7 | 64.1 |
| Top-50 | 92.1 | 74.2 | 88.0 | 80.2 |
| Top-100 | 94.4 | 82.5 | 91.2 | 89.7 |

| @Dict. Scale | HUST-OBC | | EVOBC | |
|---|---|---|---|---|
| | *Valid.* | *ZS* | *Valid.* | *ZS* |
| 7000 | 59.6 | 31.9 | 65.7 | 48.2 |
| 10000 | 73.7 | 39.3 | 79.5 | 59.8 |
| 20902 | 86.5 | 51.8 | **83.9** | **66.4** |
| 27928 | **88.3** | **54.1** | 82.2 | 64.5 |
| 47157 | 87.8 | 53.7 | 81.7 | 64.1 |

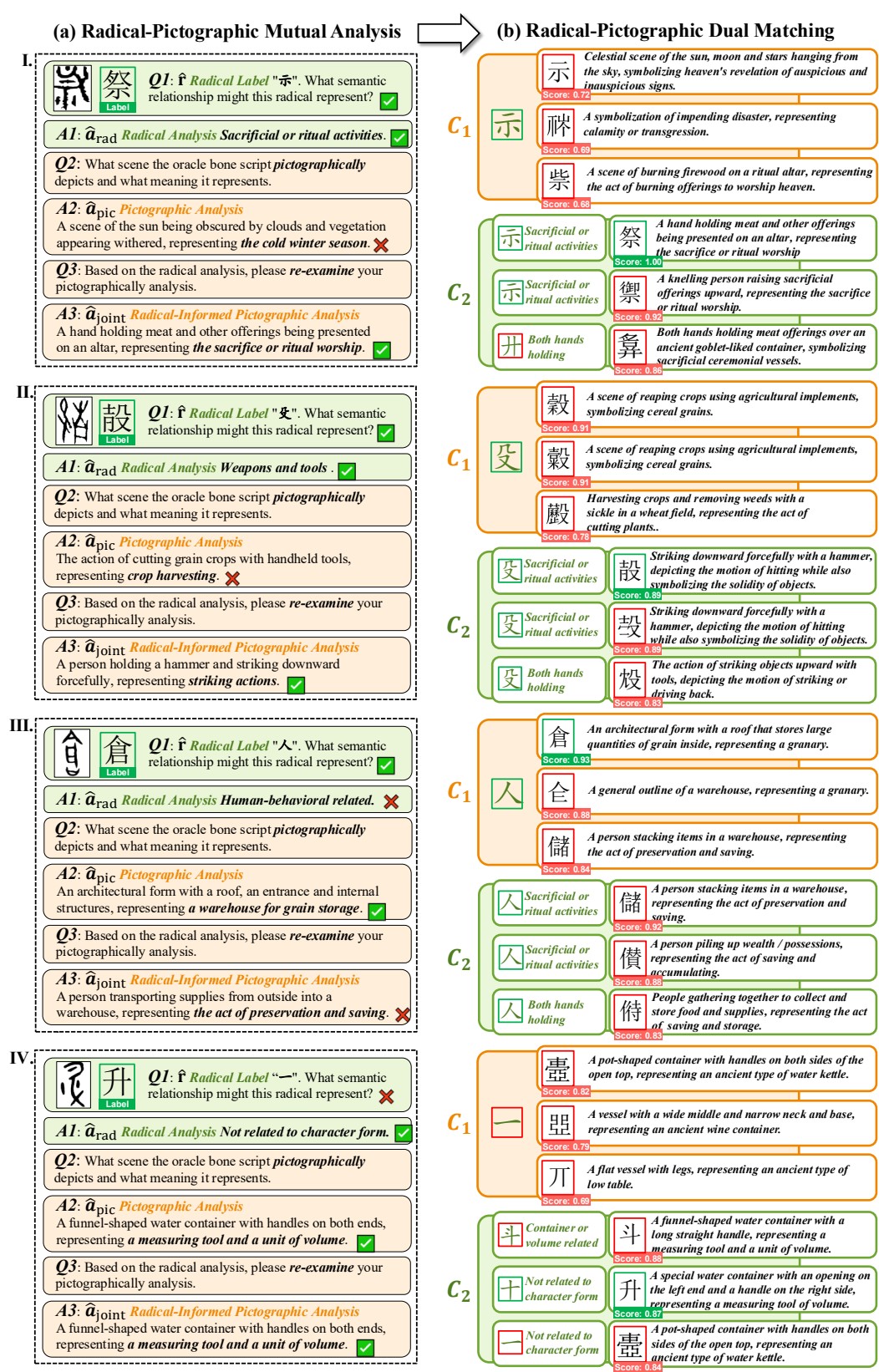

Figure 6: Visualization of Radical-Pictographic Mutual Analysis and candidate sets $C_1$ & $C_2$ in Dual Matching. Green rectangles and checkmarks indicate correct contents and results.

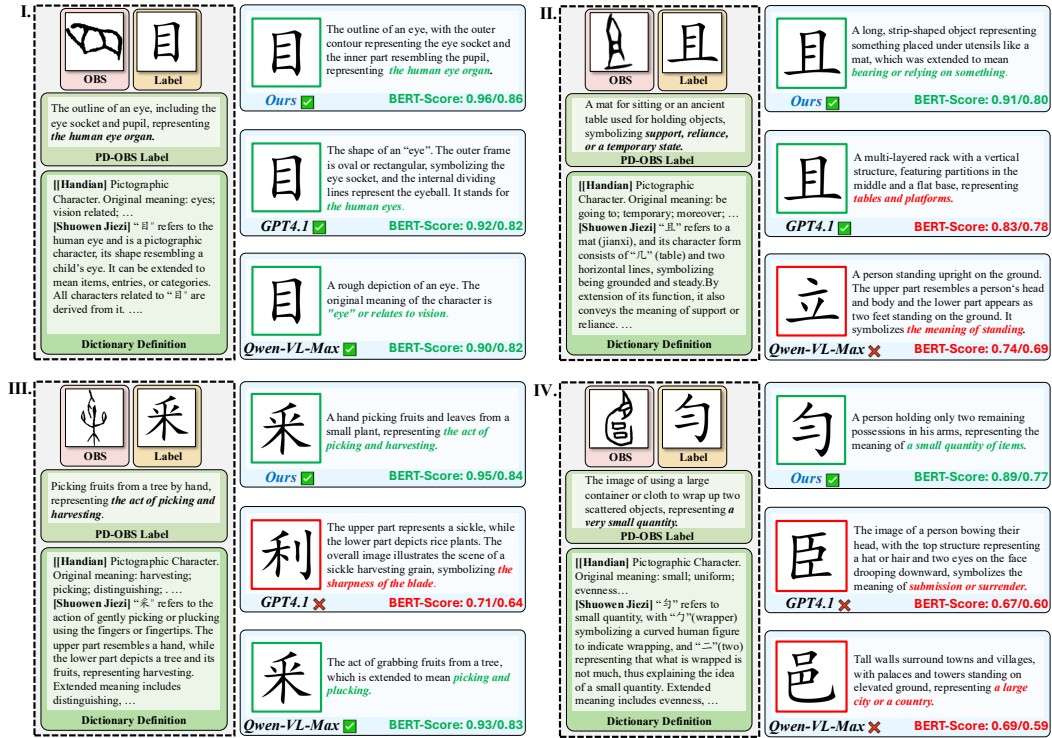

Figure 7: Visualization of interpretable outputs from LVLM-based methods. Green rectangles and texts indicate correct contents and results. The two BERT-Score values correspond to: the similarity score between the model outputs and the PD-OBS label / the similarity score between the model outputs and the authoritative dictionary definition.

### A.5.3 DICTIONARY SCALE STUDY

In practice, only a limited subset of over 100,000 Chinese characters has achieved widespread circulation and possesses well-defined meanings. Therefore, selecting an appropriate candidate dictionary for oracle bone script decipherment is crucial. Under the PD-OBS dataset setting, the matching dictionary comprises 47,157 Chinese characters documented in the Kangxi Dictionary (a Chinese dictionary).

We constructed four additional subset dictionaries: 7,000 commonly used Chinese characters, 10,000 commonly used characters including known oracle bone script decipherment results, 20,902 Unicode-supported characters, and 27,928 characters encompassing known oracle bone script decipherment results and Unicode Extension characters. We evaluated the Top-10 decipherment accuracy of our method under different dictionary scales.

As demonstrated in Table 8, larger candidate dictionaries do not necessarily yield superior results; instead, candidate dictionaries with scales ranging from 20,000 to 30,000 characters prove more suitable. This indicates a trade-off relationship between decipherment accuracy and potential recall rate. Despite a slight performance loss, we adopted the Kangxi Dictionary as the PD-OBS dictionary and reported the experimental results based on it in the main text, owing to its reliability and authority.

### A.5.4 THE IMPACT OF RADICAL RECOGNITION

To further validate the robustness of the model's multi-stage learning, we conducted experiments to investigate its performance when radical recognition errors occur, as shown in the Table 9. These results confirm that even when the radical recognition stage is incorrect, the model still succeeds in many cases. Specifically, when radical prediction errors occur, the proposed model still maintains Top-10 accuracy rates of 32.20% and 47.70% on HUST-OBC and EVOBC, respectively.

Table 9: Decipherment Top-10 accuracy in the zero-shot setting. The number of total test data is 1328.

| Dataset | Correct / Wrong Predicted Radical | Top-10 Acc. (Correct Radical) | Top-10 Acc. (Wrong Radical) |
|---|---|---|---|
| HUST-OBC | 1157 / 171 | 56.90% | 32.20% |
| EVOBC | 1461 / 155 | 65.80% | 47.70% |

### A.6 OBSERVATIONS ON MODEL OUTPUTS

#### A.6.1 MUTUAL ANALYSIS AND DUAL MATCHING

Figure 6 visualizes the detailed process of Radical-Pictographic Mutual Analysis and Radical-Pictographic Dual Matching. The left part presents the Radical-Pictographic Mutual Analysis content: Radical Analysis, Pictographic Analysis, and the Radical-Informed Pictographic Analysis. The right part displays the Top-3 candidate characters from Filtered Matching $C_1$ and Joint Matching $C_2$. The final Top-k matching results are obtained by aggregating and re-ranking the candidate characters from $C_1$ and $C_2$ according to their respective BERT-Score value Zhang et al. (2020). In the cases **I** and **II**, mutual analysis successfully rectifies errors originating from pictographic analysis, demonstrating its effectiveness and rationality. In the case **III**, the final deciphering result remained unaffected despite the introduction of unreasonable content during the radical analysis phase. This resilience comes from the error tolerance achieved through the Filtered Matching $C_1$, which is enabled only by pictographic analysis. In Case **IV**, even when radical recognition fails, the model analyzes the role of radicals in character and utilizes pictographic and mutual analysis to mitigate the impact of erroneous information, thereby obtaining the accuracy of decryption results.

#### A.6.2 INTERPRETABLE CONTENT

Figure 7 displays the interpretable outputs from three LVLM-based methods, including our method. We employed BERT-Score Zhang et al. (2020) to calculate the similarity between model outputs and ground truth annotations, evaluating the reliability of these interpretable contents as a qualitative supplement to Table 2. In Case **I**, where all models provided correct decipherment results, our method's analysis demonstrated the highest similarity with both PD-OBS annotations and authoritative Chinese dictionary definitions (Dictionary Definition). In Cases **II**, **III**, and **IV**, our model demonstrates more precise interpretability compared to models that predict correctly, and more reasonable explanations and deciphering compared to models that predict incorrectly.

### A.7 EXTENDED RESULTS

#### A.7.1 MORE VISUALIZATION RESULTS

Figure 9 visualizes additional analysis and decipherment results of our method, with the OBSD method used for comparison. The visualization results further demonstrate the effectiveness and robustness of the proposed method on multiple validation set samples and zero-shot samples. We also demonstrated a failed zero-shot case due to an error in pictographic analysis.

As shown in Figure 10, we represent decipherment outputs for previously undeciphered characters. For each character, we show the Top-4 decipherment results and the related analysis contents. We will subsequently publish all prediction results of the model for undeciphered characters, and open-source both the model and dataset to support research in this field.

#### A.7.2 CROSS-DATASET RESULTS

To validate the cross-form generalization capability and practical application value of the proposed model, we further conducted verification on the Rubbing form OBI-125 dataset Yue et al. (2022a) as shown in Table 10. Experiments demonstrate that even without training on rubbing data, the proposed model maintains competitive Top-1 accuracy and significantly outperforms method OBSD in Top-10 accuracy when confronted with rubbing data containing substantial noise. Furthermore,

Table 10: Decipherment performance Top-1 / Top-10 (in%) on cross-dataset benchmarks.

| Method | OBI-125 (rubbing form) | Oracle-50K (Validation) | Oracle-50K (Zero-shot) |
|---|---|---|---|
| OBSD | 43.1 / 49.7 | 54.8 / 62.4 | **20.5** / 25.6 |
| Ours | **46.2 / 60.3** | **55.3 / 72.5** | 19.7 / **34.8** |

| OBS Image | GT | Top-3 Result (OBSD) | Top-3 Results (Ours) | Radical Analysis (Ours) | Pictographic Character Form Analysis (Ours) |
|---|---|---|---|---|---|
| | 贞 | 贞 [贞] [页] [贞] | [贞] [贞] [禛] | Radical "贝" is related to currency, transactions, or shellfish creatures; in the character form it symbolizes shells | Using shells for divination activities, representing **the meaning of divination and observation** |
| | 未 | [未] [未] [未] | [未] [枛] [木] | Radical "木" is related to trees or plants, in the current character form it symbolizes a big tree | A tree with flourishing branches and foliage, representing **the state of a tall, vigorously growing tree.** |
| | 雨 | 舟 [局] [易] [居] | [雨] [宋] [闲] | Radical "雨" indicates rain or related weather phenomena, directly relevant to the character form. | A scene of rainwater falling from the sky, representing **rainfall** |
| | 王 | 兵 [忄] [汰] [似] | [立] [太] [恒] | Radical "立" represents a standing posture or an upright state, directly relevant to the character form. | A person standing on the ground with feet apart, expressing the meaning of **standing upright or being established.** |

Figure 8: Visualization of extended decipherment results for rubbing form data from OBI-125 dataset.

to validate the cross-dataset generalization of the proposed model, we also conducted experiments on the Oracle-50K dataset Han et al. (2020a). The proposed method achieved highly competitive Top-1 accuracy and markedly superior Top-10 accuracy. These experimental results collectively validate the effectiveness, generalization, and application value of the proposed approach.

In addition, the visualization results are shown in Figure 8 to demonstrate the deciphering process of our method for rubbing data.

## A.8 LIMITATIONS AND FUTURE WORK

The supervised fine-tuning to some extent restricts the model's generalization and reasoning capabilities. We observe that the oracle bone script dataset contains numerous characters with similar glyphs or semantics, leading the model to rely on similar character information from training labels in zero-shot testing scenarios rather than conducting thorough radical and pictographic analysis, consequently causing deviations in results. For example, when the training set involves the Chinese character "pin" composed of three "kou" radicals, and the zero-shot test contains the Chinese character "ji" composed of four "kou" radicals, the model sometimes ignores the pictographic analysis and directly outputs the label of the Chinese character "pin", because these two characters have highly similar glyphs and meanings. In addition, when an undeciphered OBS corresponds to an extremely rare modern Chinese character beyond the scope of the dictionary, our model cannot provide a definitive prediction. In such instances, the model will offer radical analysis and pictographic analysis as reference information for experts.

To address these limitations, future improvements will consider applying state-of-the-art reinforcement learning frameworks and a targeted reward function to further overcome the model's generalization constraints. Additionally, we will attempt to integrate composition-based methods to enhance the model's robustness for semantically complex yet structurally well-defined characters.

## A.9 VISUALIZATION OF THE PD-OBS DATASET

To provide a more intuitive demonstration of the proposed dataset PD-OBS, we have visualized several data cases in Figure 11. It is worth noting that, for the sake of clarity, we have translated the annotation information into English.

| Data Type | OBS Image | GT | Top-3 Result (OBSD) | Top-3 Results (Ours) | Radical Analysis (Ours) | Pictographic Character Form Analysis (Ours) |
|---|---|---|---|---|---|---|
| #Validation | | 監 | [監] [豐] [鑒] | [監] [监] [鑑] | Radical "皿" is related to basin-type vessels, in the current character form it appears as the image of **a water basin** | A person kneeling beside a basin, observing their own reflection in the water, representing the meaning of **observation and examination** |
| | | 倪 | [倪] [悗] [掜] | [倪] [倗] [观] | Radical "人" is related to human behaviors, in the current character form it symbolizes **a person** | A figure with focused gaze upon an object, representing the meaning of **observation and metaphorical reasoning** |
| | | 兮 | [兮] [兌] [仯] | [兮] [于] [六] | Radical "八" is related to paths or flow; in the character form it symbolizes **air current** | A stream of air flowing downward from above, representing **the transmission of breath or sound** |
| | | 刈 | [刈] [刊] [刘] | [刈] [割] [劖] | Radical "刂" is related to blade cutting, in the current character form it represents a **farm sickle** | A farmer harvesting wheat with a sickle, representing the meaning of **crop harvesting** |
| | | 鑄 | [詹] [屉] [底] | [鑄] [铸] [銚] | Radical "金" is related to metallic objects, in the current character form it symbolizes **metallurgy engineering** | A person holding an ancient tripod vessel (li) pouring molten metal into a mold, representing **the process of casting and metallurgy** |
| #Zero-shot | | 氾 | [氾] [已] [洰] | [氾] [洄] [沈] | Radical "水" is related to water flow, in the current character form it symbolizes **branches of water flow** | Rivers flowing into a lake water system, indicating **river inlet to the lake** |
| | | 昔 | [昔] [巷] [峕] | [昝] [昔] [晞] | Radical "日" indicates the sun, in the current character form it represents **sunshine** | A scene of meat and food under the scorching sun, representing the meaning of **dried meat** |
| | | 盗 | [盗] [盗] [㳻] | [盡] [瀺] [盆] | Radical "皿" is related to containers, in the current character form it represents **a water jar** | A hand holding a brush to clean vessels, expressing the meaning of **emptiness within containers** |
| | | 絕 | [割] [剄] [剒] | [斦] [絕] [絕] | Radical "糹" is related to silk threads, in the current character form it represents **a bundle of ropes** | A scene of cutting hemp rope with a knife, indicating a clean cut, symbolizing the meaning of **termination and severance** |
| | | 剛 | [仄] [厇] [庆] | [剛] [剛] [則] | Radical "刂" is related to blade cutting, in the current character form it represents a **sharp knife** | A scene of cutting through the interwoven network of rope fibers with a blade, representing **the qualities of rigidity and resilience** |

Figure 9: Visualization of extended decipherment results in validation and zero-shot settings.

| Oracle Bone Character | OBSD Result | Results (Ours) | Radical Analysis (Ours) | Pictographic Character Form Analysis (Ours) |
|---|---|---|---|---|
| | 洭 | [漁] [泗] [漁] [汕] | Radical "水" is related to water flow, in the current character form it represents a flowing stream | A scene of fishing with nets in a stream, representing the meaning of fishing |
| | 枠 | [春] [畱] [舂] [盥] | Radical "臼" indicates tools used for pounding rice, directly related to the character's meaning | The process of holding a pestle and mortar while pounding rice, expressing the meaning of husking grain |
| | 屮 | [洗] [湏] [浴] [淀] | Radical "水" is related to water flow, in the current character form it symbolizes a water pool | A foot being washed in water, representing foot washing |
| | 峀 | [目] [眥] [罘] [睚] | Radical "目" is related to eyes, directly related to the character's meaning | Eyes facing each other, expressing the meaning of mutual gaze or eye contact |
| | 片 | [災] [燊] [裁] [焚] | Radical "火" is related to fire and heat, in the current character form it represents burning flames | A scene of fire spreading through trees, expressing the meaning of forest fire or catastrophic blaze |
| | 虭 | [蠱] [蚤] [潫] [董] | Radical "虫" represents insect creatures, directly related to the character form | An image of a long-tailed venomous insect crawling, indicating a type of insect creature |
| | 飓 | [魒] [處] [虎] [虘] | Radical "虎" is related to tigers and beasts, in the current character form it symbolizes a pouncing tiger | A scene of a fierce tiger pouncing from the grass, indicating hidden threats and imminent danger |
| | 峓 | [韦] [弓] [㢗] [玫] | Radical "弓" is related to bows and arrors, in the current character form it symbolizes hunting with a bow | A figure grasping a bow with both hands, representing a ceremonial act before hunting |
| | 詹 | [誉] [玭] [毖] [貔] | Radical "比" is related to comparison or parallelism, unrelated to the current character's pictographic meaning | A fierce beast with bared fangs and extended claws, representing the meaning of ferocity and savagery |
| | 黽 | [黽] [龟] [電] [黿] | Radical "電" is related to turtles and reptilian animals, directly related to the character's meaning | Turtle shell patterns and turtle form, indicating a type of large turtle creature |
| | 刖 | [刖] [劇] [制] [剝] | Radical "刂" is related to blade cutting, in the current character form it represents execution tools | A criminal's legs and feet being severed, representing the ancient cruel punishment of foot amputation |
| | 聲 | [聲] [听] [聽] [磬] | Radical "耳" is related to sound and hearing, in the current character form it represents the image of an ear | A suspended bell chime, representing the resonating sound of bells |
| | 桊 | [炎] [炮] [炘] [爈] | Radical "火" is related to burning scenes, in the current character form it symbolizes fiercely burning flames | A scene of flames burning fiercely, representing vigorous fire |
| | 睫 | [異] [畐] [异] [畏] | Radical "田" is related to fields/farmland, unrelated to the current character's pictographic meaning | A complete human figure with both hands raised high, representing a bizarre form that inspires fear |
| | 綰 | [妝] [斐] [妆] [姬] | Radical "女" is related to feminine qualities, in the current character form it manifests as a kneeling woman | A woman arranging her appearance beside a bed, expressing the meaning of grooming and dressing up |

Figure 10: Visualization of extended decipherment results of undeciphered characters.

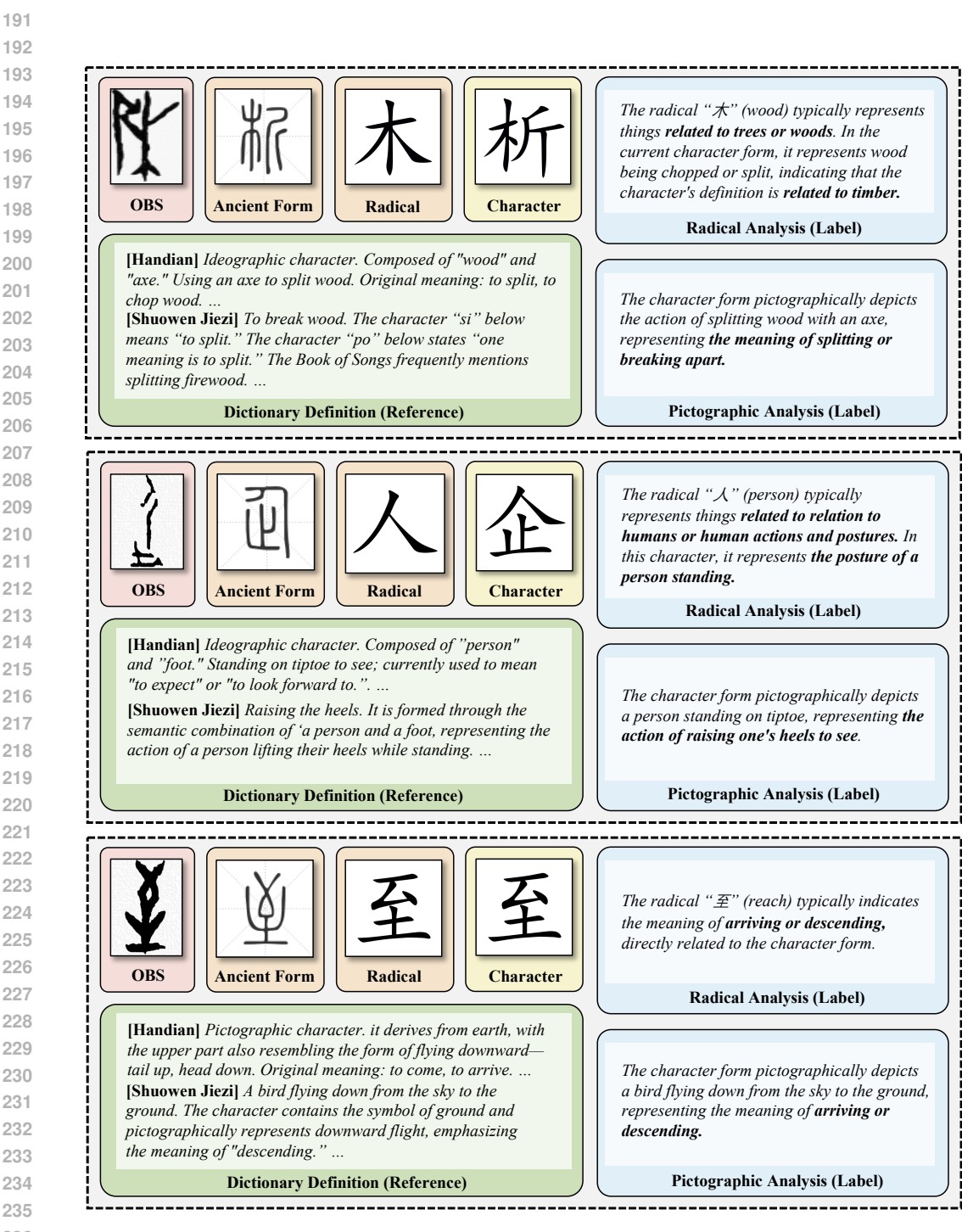

Figure 11: Visualization of the PD-OBS dataset.

