# OpenReview forum: "Interpretable Oracle Bone Script Decipherment through Radical and Pictographic Analysis with LVLMs"
_ICLR.cc/2026/Conference — Submitted to ICLR 2026_

### Official Review · Reviewer_xkzS · 2025-10-21

**Soundness:** 2
**Presentation:** 3
**Contribution:** 2
**Rating:** 4
**Confidence:** 5

**Summary:**

The paper proposes an VLM-based Oracle Bone Script decipherment. A progressive training strategy is introduced to guide the model from radical analysis to pictographic analysis, enabling rich semantic understanding. A Radical-Pictographic Dual Matching mechanism is designed to connect the decipherment results with modern Chinese characters. The authors also introduce a Pictographic Decipherment OBS Dataset, consisting of 3,173 OBS classes and 47,157 Chinese characters with glyph analysis. Experiments show that the proposed method achieves superior performance in OBS decipherment.

**Strengths:**

1. A VLM-based OBS decipherment model that combines both radical and pictographic information to generate more comprehensive interpretations.
2. The PD-OBS dataset, with its comprehensive radical and pictographic annotations, provides an enduring resource for future research in ancient Chinese character analysis and serves as a benchmark for developing interpretable AI systems in digital humanities applications.

**Weaknesses:**

1. Using pictographic structures such as radicals to interpret the meanings of oracle bone script is a common approach, and there have been many related studies.
2. PD-OBS, though large, is derived from modern Chinese and known decipherments. This may bias the model toward modern semantics rather than authentic ancient meaning reconstruction. In particular, many undeciphered oracle bone inscriptions have lost their evolutionary path and cannot correspond to modern Chinese characters.
3. Lack of visualization of PD-OBS dataset.
4. As acknowledged by the authors, LoRA fine-tuning reduces base model reasoning ability and can lead to overfitting or reliance on seen glyphs, weakening zero-shot interpretability.
5. While the approach is effective, it primarily involves clever integration and fine-tuning of existing components (LVLM, LoRA, dictionary matching) rather than introducing a fundamentally new modeling theory.
6. Since the PD-OBS dataset uses GPT-4.1 for annotation and expansion, how is annotation quality and potential LLM hallucination controlled or audited? How does the self-checking mechanism of GPT4.1 operate? Will it introduce biases?
7. Lack of cross-font decipherment effect evaluation. The current experiments mainly focus on the handprinted OBS, while most oracle bone inscriptions exist in rubbing form and contain various types of noise.
8. The performance of the overall decipherment framework is limited by the diversity of the constructed PD-OBS dictionary.
9. What is the specific construction of spatial patch merger in line: 243? Fully connected layer?
10. What mechanism is used to detect the error in A2 (Fig. 4)?
11. The logical expression in Figure 4 needs improvement. Currently, the sequence of the block diagram is difficult to understand.
12. Lack of comparison with the latest visual large models (e.g., Gemini 2.5 Pro, Claude 4.1, GPT-5) with reasoning abilities.
13. Incorrect annotation. For the HUST-OBC column in Table 1, the underlined and bolded parts at Top-10 accuracy.
14. As shown in Tab. 1, the performance improvement on validation set is not significant and even there is a decline, compared with the existing methods.
15. Table 1: What are the evaluation settings for commercial VLMs? Their rationality will significantly impact model output quality.
16. Lack of evaluation for those characters themselves are components. We call these components those with the ability to independently form words.
17. Does the model perform well when confronted with ancient characters that share *no radicals* with known ones? How does it handle rare or unique radicals?
18. The study relies mainly on HUST-OBC and EV-OBC benchmarks. Further human-expert evaluation of interpretability would strengthen the archaeological credibility.
19. Although BERTScore can capture semantic similarity, existing studies have shown that it is more suitable for evaluating the fluency of generated text rather than the accuracy of content. Further human evaluation is needed to validate its alignment with experts.

**Questions:**

See weaknesses.

---

> ### Author Response · Authors · 2025-11-25
> **Response to Reviewer xkzS (Part 1)**
>
> **Q1:** Pictographic structures (radicals) as interpretive cues are common; prior studies already pursue this line.
>
> **A1:** Thank you for the comment. We agree that using pictographic or component structures to interpret OBS is well-established in linguistic and archaeological research. Our contribution lies not in the idea itself, but in proposing a novel pipeline for translating this knowledge into an explicit, progressive LVLM-based reasoning pipeline.
>
> Concretely, we propose a progressive training strategy (radical analysis→ pictographic analysis→ mutual analysis) that enables the model to learn layered semantic information in OBS glyphs. These structured analysis outputs are then used by our Radical–Pictographic Dual Matching mechanism to retrieve the modern character from a dictionary, making the entire decipherment process interpretable.
>
> We hope our response will clear up this misunderstanding.
>
> **Q2:** PD-OBS, though large, is derived from modern Chinese and known decipherments, risking bias toward modern semantics; many undeciphered inscriptions lack a path to modern correspondences.
>
> **A2:** Thank you for raising this concern. We believe there may be a misunderstanding about the construction of PD-OBS. PD-OBS is not derived from modern Chinese and known decipherments alone. As stated in the paper (Section 3 and Supplementary material), PD-OBS consists of:
>
> - OBS images
> - Their corresponding ancient glyph forms
> - The linked modern Chinese characters
> - Fine-grained radical and pictographic analyses based on authoritative ancient dictionaries.
>
> Therefore, we have taken into account ancient characters and their meanings when training the model. Moreover, the OBS decipherment task aims to **utilize modern languages to help humanity to understand OBS [1, 2]**. Therefore, we consider **ancient meaning reconstruction not to be the primary objective** of our endeavour.
>
> Finally, when an OBS has no clear modern equivalent, our  method can output radical analysis and pictographic semantics, which remain valuable for understanding the OBS even without a definitive modern label.
>
> **Q3:**   PD-OBS lacks dataset visualizations.
>
> **A3:** Thank you for the suggestion. We present a data sample in Figure 2 of the main paper. Following your advice, a visualization of the PD-OBS dataset has been added to the revised appendix (A.9).
>
>  In addition, if you would like to see more examples, please refer to the supplementary material we uploaded.
>
> **Q4:**  As acknowledged, LoRA fine-tuning can reduce base reasoning and induce overfitting/reliance on seen glyphs, weakening zero-shot interpretability.
>
> **A4:** Thanks for the reviewer's comments. We acknowledge that this limitation exists and have stated it in the appendix A.8. It is also a common challenge shared across existing OBS decipherment approaches: composition-based method [3] may misidentify visually similar components, while diffusion-based method [1] may reconstruct a glyph into an unrelated but shape-similar training character.
>
> In our framework, LoRA introduces a practical and controlled trade-off. It allows (i) targeted infusion of radical and pictographic knowledge without modifying the LVLM backbone, (ii) progressive stage-wise adaptation aligned with our radical → pictographic → mutual analysis curriculum, and (iii) better preservation of general reasoning ability than full fine-tuning. Furthermore, our analysis-driven dual-matching mechanism alleviates LoRA-induced overfitting by relying on semantic, structured analysis rather than direct generation, thereby maintaining zero-shot robustness.
>
> As evidenced in Table 4, the  dual-matching mechanism significantly improves generalization. While the limitation cannot be completely eliminated, our method attains superior Top-10 accuracy in zero-shot, demonstrating stronger robustness compared with existing methods.
>
> ***References:***
>
> [1]Haisu Guan, et al. "Deciphering Oracle Bone Language with Diffusion Models. " *Proceedings of the 62nd Annual Meeting of* *the Association for Computational Linguistics* *(**ACL**)*, 2024.
>
> [2] Li, Caoshuo, et al. "Oraclefusion: Assisting the decipherment of oracle bone script with structurally constrained semantic typography." *Proceedings of the IEEE/CVF* *International Conference on Computer Vision* *(**ICCV**)*. 2025.
>
> [3]  Wang, Pengjie, et al. "Puzzle pieces picker: Deciphering ancient chinese characters with radical reconstruction." *International Conference on Document Analysis and Recognition*. Cham: Springer Nature Switzerland, 2024.

---

> ### Author Response · Authors · 2025-11-25
> **Response to Reviewer xkzS (Part 2)**
>
> **Q5:**  The approach chiefly integrates existing components (LVLM, LoRA, dictionary matching) rather than introducing a new modeling theory.
>
> **A5:** Thank you for the comment. We agree that our method builds on existing components such as LVLMs, LoRA and dictionary matching. However, we sincerely hope the reviewers will consider that **our core contribution lies in how these elements are organized into a progressive and interpretable decipherment pipeline**. Specifically, the radical → pictographic → mutual analysis strategy and the proposed Radical–Pictographic Dual Matching mechanism operationalize OBS domain knowledge into explicit reasoning steps, leading to strong zero-shot performance and interpretable outputs. We aspire to leverage cutting-edge artificial intelligence techniques to offer possible assistance to archaeology or historiography.
>
> **Q6**:  PD-OBS uses GPT-4.1 for annotation/expansion; annotation quality and LLM hallucination control/auditing (incl. GPT-4.1 self-checking and bias) are unspecified.
>
> **A6:** Thank you for your comments.
>
> We employ self-checking and manual correction to address potential annotation errors arising from GPT hallucinations, as demonstrated in Sec. 3. The specific approach is as follows:
>
> Each Radical & Pictographic Analysis annotation derived from GPT analysis and summarization is re-input into GPT alongside modern character forms, ancient character forms, and comprehensive dictionary analyses (including all reference information available from authoritative ancient sources likes Shuowen Jiezi, Kangxi Dictionary, and Han Dictionary). The following self-checks are then performed: a. Whether the annotation description aligns with the character form; b. Whether the annotation description conflicts with dictionary analyses or contains information not covered by them; c. Whether the annotation summarises the dictionary analysis content. All problematic annotations are regenerated and re-examined. In addition, the final output undergoes manual correction to minimize erroneous analyses.
>
> We incorporated the aforementioned specific annotation correction strategy into the Appendix A.2.5 of the revision to enhance the paper's completeness.
>
> **Q7:**  No cross-font evaluation; Emphasize handprinted OBS, whereas real inscriptions are rubbings with diverse noise.
>
> **A7:** Thank you for the comment. Following prior work such as OBSD [1] ,OracleSage [2], and PPP [3], our experiments focus on the clean, handprinted / scanned OBS datasets HUST-OBC and EVOBC, which are the established benchmarks for evaluating decipherment models. These datasets allow us to isolate and study the decipherment process itself, rather than confounding factors such as noise removal, rubbing restoration, or denoising—which constitute a separate research problem.
>
> **Q8:**  Overall performance is bounded by the diversity/coverage of the PD-OBS dictionary.
>
> **A8:** Thank you for pointing out this important limitation. We agree that the method is constrained by the scale of the dictionary. To mitigate this limitation, we selected the Kangxi Dictionary, an authoritative existing dictionary that contains a large number of Chinese characters and provides analytical content. This dictionary includes 47,157 Chinese characters, covering all commonly used characters and a substantial number of rare ones. Nevertheless, it cannot be ruled out that extremely rare characters beyond the scope of the Kangxi Dictionary may exist.
>
> Therefore, we explicitly stated the following limitation in the revision: When an undeciphered OBS corresponds to an extremely rare modern Chinese character beyond the scope of the dictionary, our model cannot provide a definitive prediction. In such instances, the model will offer radical analysis and pictographic analysis as reference information for experts.
>
> ***References:***
>
> [1]Haisu Guan, et al. "Deciphering Oracle Bone Language with Diffusion Models. " *Proceedings of the 62nd Annual Meeting of* *the Association for Computational Linguistics* *(**ACL**)*, 2024.
>
> [2] Jiang, Hanqi, et al. "Oraclesage: Towards unified visual-linguistic understanding of oracle bone scripts through cross-modal knowledge fusion." *arXiv* *preprint* *arXiv:2411.17837* (2024).
>
> [3] Wang, Pengjie, et al. "Puzzle pieces picker: Deciphering ancient chinese characters with radical reconstruction." *International Conference on Document Analysis and Recognition*. Cham: Springer Nature Switzerland, 2024.

---

> ### Author Response · Authors · 2025-11-25
> **Response to Reviewer xkzS (Part 3)**
>
> **Q9:** The construction of the spatial patch merger (line 243) is under-specified.
>
> **A9:** Thank you for the question. The spatial patch merger is designed by stacking the standard patch merger of Qwen2.5-VL multiple times, consists of RMSNorm layer and MLP.  Its role is simply to aggregate visual tokens into a compact high-level representation for radical recognation. More details in Appendix A.2.2.
>
> **Q10:** The mechanism for detecting the A2 error in Fig. 4 is under-specified.
>
> **A10:**  Thank you for the question. We would like to clarify that Fig. 4(A2) does not involve any automatic error-detection mechanism. The red mark simply indicates, for illustration purposes, that this is an example where a previous pictographic interpretation was incorrect.
>
> Our intention is to show that the proposed Radical–Pictographic Mutual Analysis helps produce more accurate and coherent pictographic interpretations, thereby reducing such errors. The figure is meant to visualize how our multi-stage analysis mitigates issues like A2, not to imply that the model explicitly detects mistakes.
>
> We have further clarified this in the revision to avoid any potential misunderstanding.
>
> **Q11:**  Figure 4’s logical flow is hard to follow; the block sequence needs improvement.
>
> **A11:** Thank you for your suggestion, and we agree with your comments. To enhance readability, we have optimised the presentation of Figure 4 in two aspects: the data flow symbols and the output annotations. Additionally, we have provided additional explanatory text for this figure in Section 4.3 of the revised manuscript, as follows:
>
> Specifically, we employ radical analysis $\hat{a}_ {\text{rad}}$ and pictographic analysis $\hat{a}_ {\text{pic}}$ as contextual information, prompting the LVLM to re-examine the pictographic analysis and generate the radical-informed pictographic analysis $\hat{a}_{\text{joint}}$, as illustrated by Q3&A3 in Figure 4. This enables the model to explicitly consider radical-implicated evidence, thereby mitigating the difficulty of directly analyzing the entire character.
>
> **Q12:** Missing comparisons with recent reasoning-capable VLMs (e.g., Gemini 2.5 Pro, Claude 4.1, GPT-5).
>
> **A12:** Thank you for the suggestion. To strengthen the comparison, we additionally evaluated GPT-5 and Gemini-2.5-Pro. Both models still achieve  lower Top-1 accuracy (<8%) and  lower interpretability scores, consistent with our earlier observations for GPT-4.1 and Qwen-VL-Max.
>
> These new results further confirm that current commercial LVLMs lack the domain knowledge and glyph–semantic reasoning needed for OBS deciphermen. We added these results to Table 1 and Table 2 in the  revision.
>
> **Table 1: Decipherment Accuracy (Top-1 / Top-10, %)**
>
> | Method         | HUST-OBC (Validation) | HUST-OBC (Zero-shot) | EVOBC (Validation) | EVOBC (Zero-shot) |
> | -------------- | --------------------- | -------------------- | ------------------ | ----------------- |
> | GPT-5          | 7.2 / 16.1            | 5.3 / 12.5           | 6.4 / 9.8          | 6.0 / 14.3        |
> | Gemini-2.5-pro | 6.3 / 13.9            | 5.1 / 10.4           | 5.0 / 8.6          | 6.2 / 15.0        |
> | Ours           | 80.6 / 87.8           | 76.3 / 81.7          | 16.8 / 53.7        | 33.3 / 64.1       |
>
> **Table 1. Interpretability performance comparison between different methods based on ROUGE-L /** **METEOR** **/ BERT-Score.**
>
> | Method         | HUST-OBC (Validation) | HUST-OBC (Zero-shot)  | EVOBC (Validation)    | EVOBC (Zero-shot)     |
> | -------------- | --------------------- | --------------------- | --------------------- | --------------------- |
> | Gemini-2.5-Pro | 0.486 / 0.501 / 0.745 | 0.421 / 0.419 / 0.712 | 0.436 / 0.447 / 0.713 | 0.529 / 0.538 / 0.749 |
> | GPT-5          | 0.572 / 0.575 / 0.783 | 0.470 / 0.468 / 0.725 | 0.520 / 0.521 / 0.755 | 0.498 / 0.501 / 0.740 |
> | Ours           | 0.914 / 0.907 / 0.946 | 0.550 / 0.525 / 0.794 | 0.887 / 0.884 / 0.937 | 0.576 / 0.586 / 0.849 |
>
> **Q13:** Table 1 contains incorrect annotation (HUST-OBC column: underlined/bolded Top-10).
>
> **A13:**  Thank you for the correction. The underline/bold annotation for the Top-10 accuracy in the HUST-OBC column of Table 1 was indeed incorrect, and we fixed this formatting error in the revised version.

---

> ### Author Response · Authors · 2025-11-25
> **Response to Reviewer xkzS (Part 4)**
>
> **Q14:** Table 1 shows limited gains—and even declines—on the validation set versus existing methods.
>
> **A14:** Thank you for the comment. This may be a misunderstanding, as  we did not clearly articulate the experimental metrics. We would like to clarify that in Oracle Bone Script decipherment research, the primary evaluation focus is the zero-shot setting  [1, 2],  which reflects a model’s ability to handle previously unseen characters—the core difficulty in real archaeological scenarios. By contrast, validation performance primarily indicates recognition capabilities for seen characters rather than decipherment ability, serving as a secondary auxiliary metric. We clarified the distinction between validation and zero-shot metrics  in the revision as outlined above,  to avoid potential misunderstandings.
>
> The slight advantage of PyGT on the validation set stems from its closed-set classification formulation, where the number of classes is fixed and known during training. This setup greatly simplifies Top-1 prediction compared with our open-set reasoning setting.  Further detailed analysis and comparisons are provided in Appendix A.3.
>
> **Q15**: Table 1 lacks evaluation settings for commercial VLMs; their rationale greatly affects output quality.
>
> **A15:** Thanks for your comments, which will enhance the completeness of our experiment.
>
> To evaluate the performance of commercial LVLMs in OBS decipherment task, we randomly select five samples from the training set as test cases following OracleSage [2], enabling these models to generate predictions through in-context learning. Each sample comprises three elements: an Oracle bone script image, its pictographic analysis, and the deciphered result. Each prediction consists of the ten most confidently predicted modern Chinese characters alongside their corresponding analysis content. To mitigate the impact of data randomness, we conducted five evaluations and calculated the average of the performances as the final results.
>
> In accordance with your recommendation, we have incorporated the aforementioned content into Section A.2.4 of the appendix in the revised draft.
>
>
> ***References:***
>
> [1]Haisu Guan, et al. "Deciphering Oracle Bone Language with Diffusion Models. " *Proceedings of the 62nd Annual Meeting of* *the Association for Computational Linguistics* *(**ACL**)*, 2024.
>
> [2] Jiang, Hanqi, et al. "Oraclesage: Towards unified visual-linguistic understanding of oracle bone scripts through cross-modal knowledge fusion." *arXiv* *preprint* *arXiv:2411.17837* (2024).

---

> ### Author Response · Authors · 2025-11-25
> **Response to Reviewer xkzS (Part 5)**
>
> **Q16:**  No evaluation for characters that are themselves components (i.e., capable of independently forming words).
>
> **A16:** Thank you for the comment. These characters that can independently form words and also serve as components are already included in both the validation and zero-shot test sets of PD-OBS. In our framework, a character whose radical equals itself does not require any special handling, because our method is not composition-based and does not rely on decomposing the character into smaller parts. The progressive analysis and dual-matching mechanism operate directly on the full glyph and its semantic cues, so such cases are naturally covered without distinction.
>
> **Q17:** Robustness to ancient characters sharing no radicals with known ones is unclear; handling of rare/unique radicals is not demonstrated.
>
> **A17:**  To include as many uncommon characters or radicals as possible, we have incorporated all authoritatively defined modern radicals from the China Ministry of Education's ‘Table of Indexing Chinese Character Components’ (GF 0011-2009),  and selected a dictionary containing up to 47,157 Chinese characters. This ensures our dictionary encompasses all common radical and characters alongside a substantial number of rare ones.
>
> Furthermore, even when extremely rare OBS characters possess radicals without modern equivalents, potentially causing radical misrecognition, our method mitigates this impact through the dual-matching mechanism, as demonstrated in the following Table 1.
>
> **Table 1: Decipherment Accuracy (Top-10, %) in Zero-Shot setting. The Number of total test data is 1328**
>
> | Dataset  | Number of Correct / Wrong Predicted Radical | Top-10 Accuracy with Correct Predicted Radical | Top-10 Accuracy with Wrong Predicted Radical |
> | -------- | ------------------------------------------- | ---------------------------------------------- | -------------------------------------------- |
> | HUST-OBC | 1157 / 171                                  | 56.90%                                         | 32.20%                                       |
> | EVOBC    | 1461 / 155                                  | 65.80%                                         | 47.70%                                       |
>
> These results confirm that even when the radical recognition stage is incorrect, the model still succeeds in many cases. Specifically, when radical prediction errors occur, the proposed model still maintains Top-10 accuracy rates of 32.20% and 47.70% on HUST-OBC and EVOBC, respectively. We included this analysis  in  Appendix of the revision.

---

> ### Author Response · Authors · 2025-11-25
> **Response to Reviewer xkzS (Part 6)**
>
> **Q18:** Heavy reliance on HUST-OBC and EV-OBC; human-expert evaluation of interpretability is needed to strengthen archaeological credibility.
>
> **A18:** In practical archaeological work, expert verification remains an indispensable component of all existing methods. However, within experimental research settings, current methods [1, 2] typically conduct zero-shot evaluations on already deciphered OBS, thereby providing reliable benchmark data for assessing model capabilities without incurring substantial expert costs, as noted in Sec.5.2. This setting provides reliable ground-truth labels, allowing  these methods to objectively assess the model’s decipherment ability without requiring additional expert verification, and we follow this established practice. The ground truth for interpretability analyses contained within the validation and zero-shot test set is sourced from authoritative dictionaries, thereby providing a degree of validation for the model's interpretability.
>
>
>
> **Q19:** BERTScore emphasizes fluency/semantic proximity rather than factual accuracy; expert human evaluation is needed to validate alignment.
>
> **A19:** Thank you for the insightful suggestion. We agree that expert human evaluation ultimately provides the strongest validation of semantic alignment, and we acknowledge its importance for future archaeological applications. However, expert evaluation is extremely costly and not yet standard practice in the experimental setting of OBS decipherment.  Current studies—including OBSD [1] and OracleSage [2]—typically rely on zero-shot evaluation over previously deciphered characters, where reliable ground-truth annotations already exist. In this setting, our goal is to evaluate whether the generated analysis content is consistent with these ground-truth descriptions.
>
> To provide a more comprehensive and widely recognized automatic evaluation, we adopt ROUGE-L [3] and METEOR[4] —metrics extensively used in image captioning [5, 6] and abstractive summarization[7] —alongside BERTScore. These metrics are designed specifically to assess content overlap and semantic consistency between generated text and reference annotations. The following table shows that the proposed method remains superior across all three metrics, supporting its interpretability.
>
> We agree that incorporating expert human assessment is a meaningful direction, and we plan to explore it in future work once resources permit.
>
> **Table 1. Interpretability performance comparison between different methods based on ROUGE-L /** **METEOR** **/ BERT-Score.**
>
> | Method         | HUST-OBC (Validation) | HUST-OBC (Zero-shot)  | EVOBC (Validation)    | EVOBC (Zero-shot)     |
> | -------------- | --------------------- | --------------------- | --------------------- | --------------------- |
> | Qwen2.5-VL-7B  | 0.355 / 0.358 / 0.694 | 0.309 / 0.301 / 0.651 | 0.341 / 0.350 / 0.683 | 0.337 / 0.348 / 0.679 |
> | Qwen-VL-Max    | 0.391 / 0.402 / 0.705 | 0.335 / 0.334 / 0.656 | 0.378 / 0.383 / 0.698 | 0.359 / 0.355 / 0.682 |
> | GPT-4.1        | 0.465 / 0.477 / 0.737 | 0.407 / 0.412 / 0.675 | 0.429 / 0.434 / 0.714 | 0.413 / 0.419 / 0.709 |
> | Gemini-2.5-Pro | 0.486 / 0.501 / 0.745 | 0.421 / 0.419 / 0.712 | 0.436 / 0.447 / 0.713 | 0.529 / 0.538 / 0.749 |
> | GPT-5          | 0.572 / 0.575 / 0.783 | 0.470 / 0.468 / 0.725 | 0.520 / 0.521 / 0.755 | 0.498 / 0.501 / 0.740 |
> | Ours           | 0.914 / 0.907 / 0.946 | 0.550 / 0.525 / 0.794 | 0.887 / 0.884 / 0.937 | 0.576 / 0.586 / 0.849 |
>
> References:
>
> [1]Haisu Guan, et al. "Deciphering Oracle Bone Language with Diffusion Models. " *Proceedings of the 62nd Annual Meeting of* *the Association for Computational Linguistics* *(**ACL**)*, 2024.
>
> [2] Jiang, Hanqi, et al. "Oraclesage: Towards unified visual-linguistic understanding of oracle bone scripts through cross-modal knowledge fusion." *arXiv* *preprint* *arXiv:2411.17837* (2024).
>
> [3] Lin, Chin-Yew. "Rouge: A package for automatic evaluation of summaries." *Text summarization branches out*. 2004.
>
> [4] Banerjee, Satanjeev, and Alon Lavie. "METEOR: An automatic metric for MT evaluation with improved correlation with human judgments." *Proceedings of the acl workshop on intrinsic and extrinsic evaluation measures for* *machine translation* *and/or summarization*. 2005.
>
> [5] Galliena, Tommaso , et al. "Embodied Image Captioning: Self-supervised Learning Agents for Spatially Coherent Image Descriptions." *Proceedings of the IEEE/CVF* *International Conference on Computer Vision**.* ICCV ,2025.
>
> [6] Wang, Peng, et al. "Ofa: Unifying architectures, tasks, and modalities through a simple sequence-to-sequence learning framework." *International conference on* *machine learning*. PMLR, 2022.
>
> [7] Liu, Yixin, et al. "BRIO: Bringing order to abstractive summarization." *arXiv* *preprint* *arXiv:2203.16804* (2022).

---

> > ### Comment · Reviewer_xkzS · 2025-11-27
> >
> > Thank you for the detailed reply from the author. At present, there are still some issues with this article:
> > 1. Why did we choose Clerical Script as the Ancient Form in the dataset? Why not other forms such as the more closed Bronze Inscriptions or Small Seal Script? What are the reasons and motivations behind this choice? The utility of the Ancient Form seems unclear in the proposed decipherment framework.
> > 2. The most fundamental issue is that the approach of deciphering oracle bone script and aligning it with modern Chinese characters has many inherent limitations.
> > 3. The experimental verification may not be sufficient. The selected two evaluation datasets, EVOBC and HUST-OBC, share highly correlated data sources. It is obvious that the experimental results are similar. It is necessary to conduct more cross-dataset validations, especially for the model's robustness. Otherwise, the generalization ability and practical application value may be limited.
> > 4. Some performance comparisons may not be fair enough. The training and testing divisions of the same dataset result in consistent Oracle images in form, but the major problem in this field lies in the diversity of forms.
> > 5. As shown in Table 1, the classification accuracy based on the traditional transformer architecture is even higher than that of the proposed algorithm. Their zero-shot experiments can be carried out through cross-dataset verification, which is an essential part of the comparison.
> >
> > Besides, please note that during the rebuttal period, the number of pages should not exceed 9. For the CR version, it can be up to 10 pages.
> >
> > At present, I am inclined to maintain the original score.

---

> > > ### Author Response · Authors · 2025-12-02
> > > **Response to Reviewer xkzS (Round 2)**
> > >
> > > **Q1:** Why choose Clerical Script as the Ancient Form in the dataset.
> > >
> > >
> > >
> > > **A1:** Thanks for your comment. Clerical Script serves as supplementary reference for constructing data with GPT-4o, aiming to enhance dataset quality without participating in the training.  We have added the following explanation in Appendix A.2.5:
> > >
> > > There is a substantial temporal gap between modern Chinese characters and OBS. To mitigate this impact, we introduce an intermediate script  based on ancient characters and use their analyses as auxiliary reference. This provides GPT-4o with richer and more reliable evidence, enabling it to produce more faithful OBS analyses. We instantiate this intermediate  script with Clerical Script for two reasons: (1) the historical emergence of Clerical Script is broadly contemporaneous with the compilation of the key reference Shuowen Jiezi, so that its glyph forms align closely with the dictionary  analyses; and (2) compared with earlier scripts (such as bronze inscriptions), Clerical Script has a substantially larger and more diverse surviving corpus, offering broader coverage of character types.
> > >
> > >
> > >
> > > **Q2:** The approach aligning OBS with modern Chinese characters, which has many inherent limitations.
> > >
> > > **A2:** Thanks for your comment. There may be a misunderstanding here. Our approach does not attempt to align OBS with modern Chinese characters. It should be noted that the matching mechanism used to search for potential modern Chinese character equivalents for OBS , but is not learnable. Instead, the learnable stages focus on radical-to-analysis and pictographic-to-analysis mappings. Thus, our approach seeks to align the glyphs with its implicit semantic information.
> > >
> > > This paradigm enables our approach to concentrate on understanding the semantic embedded within glyphs and to decipher OBS based on this information.  This also enables the pictographic signature of OBS to be more fully exploited, thereby endowing our method with enhanced zero-shot performance.
> > >
> > >
> > >
> > >
> > >
> > > **Q3, Q4 and Q5:** Cross-dataset and Cross-Form Experiments is needed.
> > >
> > > **A3, A4 and A5:** Thanks for your comments. Our current task focuses on scanned/handwritten images, without addressing data in rubbing form. We acknowledge your point that rubbing form is crucial for validating the practical application value of the method. Nevertheless, our task is more focused on deciphering rather than restoring or denoising rubbing images, adhering to established practices in current similar research such as OBSD [1], OracleFusion [2], and OracleSage  [3]. That said, to address your concerns and further validate the application value, we organized experiments as follows:
> > >
> > > We  conducted verification on the rubbing form OBI-125 dataset [4]. Experiments demonstrate that even without training on rubbing data, the proposed model maintains competitive Top-1 accuracy and significantly outperforms method OBSD in Top-10 accuracy.
> > >
> > > Furthermore, to validate the cross-dataset generalization of the proposed model, we also conducted experiments on the Oracle-50K dataset [5]. The proposed method achieved highly competitive Top-1 accuracy and markedly superior Top-10 accuracy. These experimental results collectively validate the effectiveness, generalization, and application value of our method.
> > >
> > > **Table 1: Accuracy (Top-1 / Top-10, %)**
> > >
> > > | Method | OBI-125 (rubbing-form) | Oracle-50K (Validation) | Oracle-50k (Zero-shot) |
> > > | ------ | ---------------------- | ----------------------- | ---------------------- |
> > > | OBSD   | 43.1 / 49.7            | 54.8 / 62.4             | **20.5** / 25.6        |
> > > | Ours   | **46.2 / 60.3**        | **55.3 / 72.5**         | 19.7 / **34.8**        |
> > >
> > > **Q6:** Page Limitation is 9.
> > >
> > > A6: We have further confirmed the official notification. During the discussion/rebuttal phase, the page limit will be increased to 10 pages to allow for new results/discussions.
> > >
> > >
> > >
> > > References:
> > >
> > > [1]Haisu Guan, et al. "Deciphering Oracle Bone Language with Diffusion Models. " *Proceedings of the 62nd Annual Meeting of* *the Association for Computational Linguistics* (ACL), 2024.
> > >
> > > [2] Li, Caoshuo, et al. "Oraclefusion: Assisting the decipherment of oracle bone script with structurally constrained semantic typography." *Proceedings of the IEEE/CVF* *International Conference on Computer Vision* (ICCV). 2025.
> > >
> > > [3] Jiang, Hanqi, et al. "Oraclesage: Towards unified visual-linguistic understanding of oracle bone scripts through cross-modal knowledge fusion." *arXiv* *preprint* *arXiv:2411.17837* (2024).
> > >
> > > [4] Yue, Xuebin, et al. "Dynamic dataset augmentation for deep learning-based oracle bone inscriptions recognition." *ACM Journal on Computing and Cultural Heritage* 15.4 (2022): 1-20.
> > >
> > > [5] Han, Wenhui, et al. "Self-supervised learning of orc-bert augmentator for recognizing few-shot oracle characters." *Proceedings of the Asian Conference on Computer Vision*. 2020.

---

### Official Review · Reviewer_DqfC · 2025-10-29

**Soundness:** 3
**Presentation:** 2
**Contribution:** 3
**Rating:** 4
**Confidence:** 5

**Summary:**

This paper utilizes LVLMs to establish the relationship between the radicals, pictographs, and meanings in Oracle Bone Script and explains the deciphering process. The paper designs a progressive training strategy that guides the model from radical analysis to pictograph analysis, ultimately achieving an interpretable deciphering process. Furthermore, the paper introduces the PD-OBS dataset, which includes Chinese character radicals and detailed glyph analysis. Experiments on publicly available benchmark tests demonstrate that the proposed method performs well in terms of deciphering ability and interpretability.

**Strengths:**

1.	The use of LVLMs, combined with radical and pictograph analysis, provides excellent interpretability for deciphering Oracle Bone Script.
2.	A progressive training strategy is proposed, gradually transitioning from the radical features of Oracle Bone Script to pictograph analysis, ultimately obtaining a more comprehensive understanding of glyph semantics.
3.	The PD-OBS dataset is introduced, containing detailed pictograph analysis annotations for 3,173 types of Oracle Bone Script and 47,157 types of Chinese characters.
4.	The experiments in the paper are thorough, and the visualization analysis is well done.

**Weaknesses:**

1.	The paper claims to be the first method to explain the deciphering process through pictograph analysis. However, in reality, both OracleSage and OracleFusion have used Oracle Bone Script components and pictograph information for deciphering. I believe the main contribution of this paper is the construction of a new pipeline that integrates radical and pictograph information for deciphering, rather than being the first to propose it. The term "first" may be misleading and could be off-putting.
2.	Similarly, PD-OBS is not the first dataset annotated with detailed radical and pictograph analysis. Both the OracleSem dataset introduced in OracleSage and the RMOBS dataset proposed in OracleFusion contain similar annotations. The PD-OBS dataset proposed in this paper can be said to be more detailed and larger in scale, but it cannot be called the "first."
3.	The paper claims to achieve state-of-the-art (SOTA) deciphering accuracy, robust zero-shot capability, and interpretability. However, based on Table 1, the method proposed in this paper is not SOTA in Oracle Bone Script recognition and zero-shot deciphering. It is comparable to OBSD in the Top-1 zero-shot task. Therefore, the paper should state that it strikes a good balance across these three aspects, rather than claiming it to be state-of-the-art in all areas.

4.	The components of Oracle Bone Script and the radicals of modern Chinese characters do not necessarily correspond one-to-one. For some Oracle Bone Script characters, the proposed method may not solve the problem effectively. In lines 156-157, the authors mention that the OBSD method has the drawback of producing unpredictable outputs. However, Oracle Bone Script itself may not have a direct correspondence with modern Chinese characters for every glyph. This is actually an advantage of OBSD, as it is not limited to a strict one-to-one correspondence between components and radicals. Therefore, I suggest that the paper should only emphasize OBSD’s output instability and lack of interpretability.
5.	In the OBSD method, the OCR engine provided can recognize nearly 90,000 types of Chinese characters. However, I noticed that the PD-OBS dataset in this paper contains 47,157 types of modern Chinese characters, and in the supplementary materials, I found that some characters lack "reference" annotations. This suggests that the deciphering scope of the proposed method may be limited to these 47,157 characters, which is narrower than the OBSD method. This is a weakness of the paper.
6.	The radical analysis method proposed in the paper seems to only recognize a single radical in Oracle Bone Script, whereas both Oracle Bone Script and modern Chinese characters may consist of multiple radicals. OracleFusion analyzes each component of Oracle Bone Script, and in comparison, the method proposed in this paper has limitations.
7.	There is an error in Figure 1, where two (b) labels appear and no (C) is shown. The reference to the EVOBC dataset is also incorrect; the author should be Guan et al., not Wang et al.

**Questions:**

I am inclined to accept your paper and increase the score, but I would need you to address each point raised in the Weaknesses section with explanations and clarifications, and remove any absolute or inappropriate wording. Additionally, you should pay attention to the various references and figure details in the paper, as these adjustments will significantly improve the overall quality of the article.

---

> ### Author Response · Authors · 2025-11-25
> **Response to Reviewer DqfC (Part 1)**
>
> We sincerely thank you for your meticulous reading of our paper, appendix, and even supplementary materials , and consider your review to be both conscientious and rigorous.  Your comments have very precisely pointed out some weaknesses and errors, which are crucial for us to improve the quality of the paper.
>
> **Q1:** The “first” claim is overstated: prior work (OracleSage, OracleFusion) already leverages Oracle Bone Script components and pictograph information for deciphering. The contribution is better framed as a new pipeline integrating radical and pictographic cues, rather than a first proposal.
>
> **A1:** Thank you for this valuable feedback, which helps us refine our contribution statement to be more rigorous. Our original intention was to express that our method is the first to explicitly combine both radical analysis and pictographic semantic analysis to explain the decipherment process. We used the term “glyph analysis” in a broad sense to cover both aspects, which unfortunately may have caused ambiguity.
>
> After re-examining OracleSage and comparing it carefully with our proposed approach, we fully agree with you that the phrase “first method to explain the decipherment process using glyph analysis” is not sufficiently precise and could be misleading.
>
> Following your suggestion, we have revised Contribution 1 to the following more accurate formulation:
>
> “We propose an LVLM-based decipherment framework to bridge the gap between glyphs and meanings in OBS, integrating radical and pictographic analyses for OBS decipherment and explicating the decipherment process.”
>
> **Q2:** PD-OBS is not the first dataset with detailed radical and pictograph analysis. OracleSem (OracleSage) and RMOBS (OracleFusion) already include similar annotations. PD-OBS is more detailed and larger in scale, but the “first” claim is inaccurate.
>
> **A2:** Thanks for your correction again. We revised Contribution 2 in the revision as : We propose the PD-OBS dataset containing multiple OBS images and related ancient and modern characters, annotated with detailed radical and pictographic analysis from authoritative classical dictionaries, providing a well-structured benchmark for OBS research.
>
> **Q3:** The SOTA claim is overstated: Table 1 shows the method is not SOTA on Oracle Bone Script recognition or zero-shot decipherment and is comparable to OBSD on Top-1 zero-shot. The contribution is better framed as striking a strong balance among deciphering accuracy, zero-shot capability, and interpretability, rather than SOTA across all aspects.
>
> **A3:**  Thank you for your suggestions, which will help enhance the rigour of this paper. We shall modify the contribution 4 to: “Our method achieves competitive performance on both oracle bone recognition and decipherment tasks, significantly outperforms existing approaches in zero-shot Top-10 accuracy, and additionally offers fine-grained interpretability. This well-balanced combination of accuracy, zero-shot generalization, and interpretability makes our approach highly promising for applications in related fields.”
>
> **Q4:** Oracle Bone Script components and modern Chinese radicals do not correspond one-to-one. For certain characters, the proposed method may be ineffective. Although lines 156–157 characterize OBSD as producing unpredictable outputs, the absence of strict one-to-one correspondence is actually an advantage of OBSD, which is not constrained by component–radical mapping. The critique should focus on OBSD’s output instability and lack of interpretability.
>
> **A4:** Thank you for your suggestion. We fully agree that the unpredictable output of OBSD holds positive significance, particularly when encountering OBS without corresponding modern Chinese characters. Therefore, this characteristic should not be regarded as a defect. In accordance with your recommendation, we shall revise the description of OBSD as follows:
>
> The unpredictable output of OBSD enables its predictions to transcend dictionary constraints, yet it still suffers from instability and a lack of interpretability.

---

> ### Author Response · Authors · 2025-11-25
> **Response to Reviewer DqfC (Part 2)**
>
> **Q5:** OBSD’s OCR engine covers ~90,000 Chinese characters, whereas PD-OBS includes 47,157 modern characters, and some entries in the supplementary material lack “reference” annotations. This indicates the proposed method’s deciphering scope is effectively limited to these 47,157 characters, narrower than OBSD, constituting a weakness.
>
> **A5:** Thank you for pointing out this important limitation. We agree that the method is constrained by the scale of the dictionary. To mitigate this limitation, we selected the Kangxi Dictionary, an authoritative existing dictionary that contains a large number of Chinese characters and provides analytical content. This dictionary includes 47,157 Chinese characters, covering all commonly used characters and a substantial number of rare ones. Nevertheless, it cannot be ruled out that extremely rare characters beyond the scope of the Kangxi Dictionary may exist.
>
> Therefore, we explicitly stated the following limitation in the revision: When an undeciphered OBS corresponds to an extremely rare modern Chinese character beyond the scope of the dictionary, our model cannot provide a definitive prediction. In such instances, the model will offer radical analysis and pictographic analysis as reference information for experts.
>
> The missing character references were due to an oversight during the compilation of supplementary materials rather than dictionary coverage limitations, which we have now corrected.
>
> **Q6:** The proposed radical analysis appears limited to recognizing a single radical per Oracle Bone Script character, whereas both Oracle Bone Script and modern Chinese characters can comprise multiple radicals. OracleFusion analyzes each component, highlighting a limitation of the proposed method.
>
> **A6:**   Thank you for your insightful comment. In Chinese, the term "radical" encompasses two distinct concepts: radical components (pianpang) and the indexing radical (bushou). Radical components denote the constituent elements of compound Chinese characters, whereas the indexing radical refers to the single representative component used for character retrieval in dictionary compilation. In our work, "radical" specifically refers to the unique indexing radical of each character.
>
> Considering the importance of the indexing radical (bushou) in character retrieval due to their typical semantic function, we adopt the indexing radical (bushou) and its analysis as the key conditions for dictionary matching.
>
> However, we fully acknowledge your perspective that analyzing both the indexing radical (bushou) and radical components (pianpang) may constitute a more comprehensive and effective approach, which we will explore in future work. Additionally, we explained radical components (pianpang) and the indexing radical (bushou) in the Appendix of the revision to avoid potential misunderstandings.
>
> **Q7:** Figure 1 contains a labeling error: two panels are marked “(b)” and no “(c)” is shown. The citation for the EVOBC dataset is also incorrect; the author should be Guan et al., not Wang et al..
>
> **A7:** Thank you for pointing out these errors. We have corrected the duplicate (b) label in Figure 1 and revised the citation for the EVOBC dataset in the revision. We have also rechecked the entire text to address the potential errors.
>
> **Q8:** Inclined to accept and raise the score, contingent on addressing each item in the *Weaknesses* section with clear explanations and clarifications, and removing absolute or inappropriate wording. Please also correct references and figure details; these adjustments will substantially improve overall quality.
>
> **A8:** We are most grateful for your thoughtful assessment and consideration of raising the score.
>
> We have revised the manuscript, removing any absolute or potentially inappropriate phrasing, and thoroughly review all references, figure captions, citation details, and presentation issues to ensure the revised version is clear, rigorous, and accurate. We hope our response will solve your concerns.

---

> ### Comment · Reviewer_DqfC · 2025-11-26
> **Official Comment by Reviewer DqfC**
>
> I appreciate the detailed response. Most of my concerns have been addressed. I decide to raise my rating.

---

### Official Review · Reviewer_u3mu · 2025-10-30

**Soundness:** 3
**Presentation:** 3
**Contribution:** 3
**Rating:** 6
**Confidence:** 3

**Summary:**

This paper presents a framework for Oracle Bone Script (OBS) decipherment based on Large Vision-Language Models (LVLMs). The core of the method is a progressive training strategy that moves from radical analysis to pictographic analysis and then to mutual analysis. Instead of direct classification, the framework retrieves the corresponding modern Chinese character from a dictionary using a proposed "Radical-Pictographic Dual Matching" mechanism. To support this approach, the authors have also constructed the PD-OBS dataset. Experiments conducted on the HUST-OBC and EVOBC benchmarks demonstrate that the method achieves high Top-10 accuracy and shows good performance in zero-shot settings, along with enhanced interpretability.

**Strengths:**

1) An Interpretable Pipeline Based on OBS Structure: The paper introduces a well-structured and interpretable pipeline for decipherment. The methodology is built upon a progressive, three-stage analysis that begins with radical analysis, proceeds to pictographic analysis, and culminates in a mutual analysis stage. This approach is explicitly defined and logically motivated, improving the semantic alignment between ancient glyphs and their modern meanings. The proposed methodology is straightforward and logically presented.
2) Construction of the PD-OBS Dataset with Analysis Annotations: The paper contributes the PD-OBS dataset, which includes detailed analytical annotations. The composition of this dataset is clearly described. The "data engine" employed to generate these annotations integrates authoritative sources with GPT-4.1, followed by both automated self-checking and manual correction steps.

**Weaknesses:**

1) Limitations in Performance and Incomprehensive Comparisons:
(1.1) The reported performance, while notable in certain metrics, reveals some limitations. For instance, in the validation setting on the HUST-OBC and EVOBC dataset, the model's Top-1 accuracy is below that of the PyGT baseline. And in the zero-shot setting on the HUST-OBC dataset, the model's Top-1 accuracy is slightly below that of the OBSD baseline, with its primary advantage appearing in the Top-10 results (Table 1). This suggests that while the method is effective at generating a list of plausible candidates, its precision in identifying the single best answer at the top rank could be improved.
(1.2) The comparison with commercial Large Vision-Language Models (LVLMs) is not sufficiently comprehensive. The authors note that the tested models (GPT-4.1 and Qwen-VL-Max) yield poor results (<6% Top-1 accuracy). However, the comparison does not include several of the most recent and powerful state-of-the-art multimodal models, such as GPT5, Gemini 2.5 Pro, claude-opus-4-1, grok-4, or others that are considered current leaders in the field. Benchmarking against these would provide a more convincing measure of the proposed method's capabilities.
(1.3) The exclusion of several recent methods on the grounds of "dataset inconsistency" weakens the experimental evaluation. To properly situate the contributions of this work, it is important to include direct comparisons with these relevant baselines on a consistent experimental setup. The authors should therefore re-evaluate these methods or provide a more compelling justification for their omission.
2) Lack of Robustness Analysis for the Multi-Stage Pipeline: The paper proposes a multi-stage pipeline, which inherently raises concerns about error propagation. The performance of later stages likely depends heavily on the accuracy of earlier ones. For instance, it is unclear how the final decipherment would be affected if the initial "Radical Recognition" stage yields an incorrect prediction. The paper does not include ablation studies or a sensitivity analysis to quantify the impact of such upstream errors. This omission makes it difficult to assess the model's overall robustness, especially when dealing with ambiguous or noisy inputs that could lead to failures in the initial stages.
3) Unsubstantiated Qualitative Comparisons in Figure 1: The comparative chart on the right side of Figure 1, which rates different paradigms as "Poor," "Medium," or "Good," lacks rigor. The paper fails to provide any objective criteria, quantitative thresholds, or a clear rubric for what constitutes each of these qualitative labels. Without a defined standard, this assessment appears subjective and potentially misleading. For such a comparison to be meaningful, it should be supported by either direct numerical evidence or a detailed set of criteria that justifies the assigned ratings.

**Questions:**

Please provide an analysis and explanation of "Weaknesses".

---

> ### Author Response · Authors · 2025-11-25
> **Response to Reviewer u3mu (Part 1)**
>
> **Q1.1:** The reported performance, while notable on several metrics, exhibits limitations: in the validation setting, Top-1 accuracy trails the PyGT baseline; in the zero-shot setting on HUST-OBC, Top-1 is slightly below OBSD. This pattern indicates strong candidate recall but insufficient precision at rank 1.
>
> **A1.1:** Thank you for the comment. This may be a misunderstanding, as  we did not clearly articulate the experimental metrics. We would like to clarify that in Oracle Bone Script decipherment research, the primary evaluation focus is the zero-shot setting  [1, 2],  which reflects a model’s ability to handle previously unseen characters—the core difficulty in real archaeological scenarios. By contrast, validation performance primarily indicates recognition capabilities for seen characters rather than decipherment ability, serving as a secondary auxiliary metric.
>
> The slight advantage of PyGT on the validation set stems from its closed-set classification formulation, where the number of classes is fixed and known during training. This setup greatly simplifies Top-1 prediction compared with our open-set reasoning framework. A similar phenomenon occurs with OBSD in zero-shot Top-1 accuracy: its diffusion-based generation tends to produce a single high-confidence prediction, whereas our method is designed to provide interpretable  reasoning and a ranked list of plausible candidates, which is more aligned with the decipherment workflow. Further detailed analysis and comparisons have provided in Appendix A.3.
>
> We clarified the distinction between validation and zero-shot metrics  in the revision as outlined above,  to avoid potential misunderstandings.
>
> **Q1.2:**  Commercial LVLM comparisons are under-scoped: only GPT-4.1 and Qwen-VL-Max are tested, excluding GPT-5, Gemini 2.5 Pro, Claude-Opus-4-1, Grok-4, and other current leaders, limiting the persuasiveness of the evaluation.
>
> **A1.2:** Thank you for the suggestion. To strengthen the comparison, we additionally evaluated GPT-5 and Gemini-2.5-Pro on HUST-OBC and EVOBC. As shown below, both models still achieve  lower Top-1 accuracy (<8%) and  lower interpretability scores, consistent with our earlier observations for GPT-4.1 and Qwen-VL-Max.
>
> These new results further confirm that current commercial LVLMs lack the domain knowledge and glyph–semantic reasoning needed for OBS decipherment. We added these results to Table 1 and Table 2 in the  revision.
>
> **Table 1: Decipherment Accuracy (Top-1 / Top-10, %)**
>
> | Method         | HUST-OBC (Validation) | HUST-OBC (Zero-shot) | EVOBC (Validation) | EVOBC (Zero-shot) |
> | -------------- | --------------------- | -------------------- | ------------------ | ----------------- |
> | GPT-5          | 7.2 / 16.1            | 5.3 / 12.5           | 6.4 / 9.8          | 6.0 / 14.3        |
> | Gemini-2.5-pro | 6.3 / 13.9            | 5.1 / 10.4           | 5.0 / 8.6          | 6.2 / 15.0        |
> | Ours           | 80.6 / 87.8           | 76.3 / 81.7          | 16.8 / 53.7        | 33.3 / 64.1       |
>
> **Table 1. Interpretability performance comparison between different methods based on ROUGE-L /** **METEOR** **/ BERT-Score.**
>
> | Method         | HUST-OBC (Validation) | HUST-OBC (Zero-shot)  | EVOBC (Validation)    | EVOBC (Zero-shot)     |
> | -------------- | --------------------- | --------------------- | --------------------- | --------------------- |
> | Gemini-2.5-Pro | 0.486 / 0.501 / 0.745 | 0.421 / 0.419 / 0.712 | 0.436 / 0.447 / 0.713 | 0.529 / 0.538 / 0.749 |
> | GPT-5          | 0.572 / 0.575 / 0.783 | 0.470 / 0.468 / 0.725 | 0.520 / 0.521 / 0.755 | 0.498 / 0.501 / 0.740 |
> | Ours           | 0.914 / 0.907 / 0.946 | 0.550 / 0.525 / 0.794 | 0.887 / 0.884 / 0.937 | 0.576 / 0.586 / 0.849 |
>
> References:
>
> [1]Haisu Guan, et al. "Deciphering Oracle Bone Language with Diffusion Models" *Proceedings of the 62nd Annual Meeting of* *the Association for Computational Linguistics* *(**ACL**)*, 2024.
>
> [2] Li, Caoshuo, et al. "Oraclefusion: Assisting the decipherment of oracle bone script with structurally constrained semantic typography." *Proceedings of the IEEE/CVF* *International Conference on Computer Vision* *(**ICCV**)*. 2025.

---

> ### Author Response · Authors · 2025-11-25
> **Response to Reviewer u3mu (Part 2)**
>
> **Q1.3:** Excluding recent methods on the basis of “dataset inconsistency” weakens the evaluation. Direct comparisons under a consistent  setup are needed, or a more compelling justification should be provided.
>
> **A1.3:** Thank you for your suggestion. We shall endeavour to compare existing methods to the greatest extent possible. We note that PPP [1] was only open-sourced in September, and therefore supplemented the experiments as follows which is added to Table 1 in the revision. Composition-based PPP produce singular prediction output but struggles to generate multiple candidate sets and still exhibits limitations in zero-shot performance significantly underperforming our approach.
>
> **Table 1: Decipherment Accuracy (Top-1 / Top-10, %)**
>
> | Method | Validation HUST-OBC | Validation EVOBC | Zero-shot HUST-OBC | Zero-shot EVOBC |
> | ------ | ------------------- | ---------------- | ------------------ | --------------- |
> | PPP    | 76.8 / -            | 72.4 / -         | 13.6 / -           | 19.1 / -        |
> | Ours   | 80.6 / 87.8         | 76.3 / 81.7      | 16.8 / 53.7        | 33.3 / 64.1     |
>
> Additionally, there are some recent methods such as OracleSage [2] and Oracle-Fusion [3]. As their source code remains unpublished and certain experimental details are difficult to obtain, we have not yet succeeded in fully reproducing these works. This makes it difficult for us to include them as comparison methods. We declared this in the appendix A.3 of the revised manuscript.
>
> **Q2:** Lack of robustness analysis for the multi-stage pipeline: the design invites error propagation, with later stages depending on earlier ones (e.g., final decipherment degrades when the initial Radical Recognition stage is incorrect). The paper omits ablations and sensitivity analysis to quantify the impact of upstream errors, limiting assessment under ambiguous or noisy inputs.
>
> **A2:** Thank you for raising this concern. We clarify that the proposed multi-stage pipeline does not operate as a strictly sequential dependency chain. Instead, each stage contributes complementary information, and the final prediction is determined by the Radical–Pictographic Dual Matching mechanism (Algorithm 1), which jointly considers:
>
> - ***C₁***: candidates filtered by radical recognition
> - ***C₂***: candidates scored from full pictographic and semantic analysis
>
> Thus, the final decipherment does not rely exclusively on the correctness of radical recognition. This seems to be a misunderstanding.
>
> To demonstrate the robustness, we have provide examples in  Appendix A.6.1 Figure 6 (Cases III and IV), where the radical recognition or radical analysis is incorrect, yet the model still produces the correct final decipherment.
>
> We further conducted an additional robustness analysis by examining cases where the radical recognition stage fails. The results show that the model still retains substantial decipherment ability:
>
> **Table 2: Decipherment Accuracy (Top-10, %) in Zero-Shot setting. The Number of total test data is 1328.**
>
> | Dataset  | Number of Correct / Wrong Predicted Radical | Top-10 Accuracy with Correct Predicted Radical | Top-10 Accuracy with Wrong Predicted Radical |
> | :------: | :-----------------------------------------: | :--------------------------------------------: | :------------------------------------------: |
> | HUST-OBC |                 1157 / 171                  |                     56.90%                     |                    32.20%                    |
> |  EVOBC   |                 1461 / 155                  |                     65.80%                     |                    47.70%                    |
>
> These results confirm that even when the radical recognition stage is incorrect, the model still succeeds in many cases. Specifically, when radical prediction errors occur, the proposed model still maintains Top-10 accuracy rates of 32.20% and 47.70% on HUST-OBC and EVOBC, respectively. We included this analysis  in  Appendix A.5.4 of the revision.
>
> ***References:***
>
> [1]  Wang, Pengjie, et al. "Puzzle pieces picker: Deciphering ancient chinese characters with radical reconstruction." *International Conference on Document Analysis and Recognition*. Cham: Springer Nature Switzerland, 2024.
>
> [2] Jiang, Hanqi, et al. "Oraclesage: Towards unified visual-linguistic understanding of oracle bone scripts through cross-modal knowledge fusion." *arXiv* *preprint* *arXiv:2411.17837* (2024).
>
> [3] Li, Caoshuo, et al. "Oraclefusion: Assisting the decipherment of oracle bone script with structurally constrained semantic typography." *Proceedings of the IEEE/CVF* *International Conference on Computer Vision* *(**ICCV**)*. 2025.

---

> ### Author Response · Authors · 2025-11-25
> **Response to Reviewer u3mu (Part 3)**
>
> **Q3:**  Unsubstantiated qualitative comparisons in Figure 1: the “Poor/Medium/Good” ratings lack rigor, with no objective criteria, quantitative thresholds, or defined rubric. Absent a standard, the assessment is subjective and potentially misleading; meaningful comparison requires numerical evidence or a detailed criterion set justifying each label.
>
> **A3:** Thank you for pointing this out. We agree that the “Poor / Medium / Good” labels in Figure 1 may appear subjective in the current form. Our intention was only to provide a visual summary of the quantitative trends reported in Tables 1 and 2 rather than to introduce new evaluation standards. However, as you suggested, to ensure a more objective presentation, we added an explanatory note to the caption of Figure 1:  Qualitative summary of three paradigms for OBS decipherment. “Poor / Medium / Good” reflects trends in Tables 1: Accuracy and Zero-shot correspond to the validation and zero-shot metrics.  Interpretability is defined by the level of intermediate analysis: no interpretable cues (Poor), coarse structural hints (Medium), and fine-grained reasoning (Good).

---

> > ### Comment · Reviewer_u3mu · 2025-11-26
> >
> > Thank you for your efforts in preparing the rebuttal, which has satisfactorily addressed all of my concerns. Therefore, I will keep my positive score unchanged.

---

### Official Review · Reviewer_27eb · 2025-10-31

**Soundness:** 2
**Presentation:** 3
**Contribution:** 2
**Rating:** 4
**Confidence:** 4

**Summary:**

The paper addresses the issue of the gap between glyphs and semantics in the field of oracle bone script decipherment and the lack of interpretability. It proposes an interpretable framework for deciphering oracle bone script based on Large Vision-Language Models (LVLMs). Through a progressive training strategy, the model is guided to gradually understand oracle bone script, enabling its decipherment. Additionally, an ideographic decipherment oracle bone script dataset is constructed, which includes 3,173 oracle bone script glyph classes and 47,157 characters from different dynasties. The method is validated on two benchmark datasets, HUST-OBC and EVOBC, demonstrating the effectiveness of the approach.

**Strengths:**

1.	The progressive training design proposed in the paper aligns with the cognitive logic of oracle bone script, following a training sequence from radicals to pictograms and then to interactions. This sequence is consistent with the evolutionary rule of oracle bone script, which progresses from shape construction to semantic expression, thus avoiding semantic confusion caused by directly analyzing complete characters.

2.	The constructed PD-OBS dataset includes detailed radical and pictogram analysis, providing high-quality benchmark data for future AI methods for oracle bone script.

3.	The overall writing of the paper is clear, the method is well-explained, and the illustrations clearly demonstrate the overall process of the method, making it highly understandable.

**Weaknesses:**

1.	The paper points out that supervised fine-tuning restricts the model's generalization and inference capabilities to some extent. However, in the task of oracle bone script decipherment, there are many characters with similar shapes or meanings. In zero-shot testing scenarios, the model might rely on information from similar characters in the training labels, rather than performing a complete analysis of radicals and pictograms.

2.	The method proposed in the paper is primarily based on large models and uses progressive training to guide the model in establishing associations between shapes and semantics in stages, addressing the issue of insufficient domain knowledge for oracle bone script in large models. However, this mechanism seems more like a customization for the oracle bone script decipherment task, with overall innovation being somewhat limited.

3.	Current methods for undeciphered oracle bone script mainly provide potential reference content, relying on human experts for further verification. The model's Top-k outputs offer possible references, but the result depends on dictionary matching, which still requires manual intervention in practical archaeological applications.

4.	The details of the method need further refinement. Some formulas lack proper punctuation, key variables are not defined, and some hyperparameters are not explained with their specific values or settings.

**Questions:**

1.	The paper highlights some hyperparameters, such as how the loss function weights are set. Was hyperparameter sensitivity analysis conducted?

2.	In the radical-pictogram dual matching mechanism, semantic similarity mainly relies on BERT-Score. However, the pre-training data for BERT mainly consists of modern text, with limited semantic information related to oracle bone script. How can it be ensured that semantic differences between ancient and modern texts do not cause biases in the similarity judgment?

3.	In Section 4.2, the paper mentions the design space patch merger as a visual adapter, which is used to downsample visual embeddings and obtain representative feature vectors suitable for classification tasks. What is the specific process here? How can it be ensured that the obtained feature vectors fully consider the structural features of the oracle bone script, especially in complex script structures, and how can the downsampling process avoid losing detailed and structural information?

---

> ### Author Response · Authors · 2025-11-25
> **Response to Reviewer 27eb (Part 1)**
>
> **Q1:** Supervised fine-tuning may restrict generalization and inference. In zero-shot testing, the model might rely on information from similar characters in the training labels, rather than performing a complete analysis of radicals and pictograms.
>
> **A1:** Thanks for the reviewer's comments. We acknowledge that this is indeed a limitation of our methodology and have declared it in the Sec A.8 of Appendix. However, it should be stated that  this is a common limitation across existing approaches and one of the most challenging issue in OBS decipherment task.
>
> For instance, composition-based method [1] may misidentify character components as shape-similar components encountered during training, while diffusion-based method [2] may reconstruct a character as one with a similar shape from the training set. In contrast to these methods that directly predict modern-form components or characters through matching or generation, our approach employs multi-dimensional fine-grained analysis—namely radical analysis, pictographic analysis, and mutual analysis—and utilises these fine-grained insights for matching. Furthermore, characters or components with similar shapes may possess similar semantic information due to the pictographic nature of OBS. Consequently, matching based on analysis rich in semantic information is a more robust approach.
>
> Therefore, our proposed dual-matching strategy mitigates this issue to some extent. Table 4 in our ablation experiments validates the effectiveness of the strategy in enhancing model's generalization. Although such limitations cannot be entirely eliminated, our model achieves superior generalization compared to other approaches, particularly demonstrating robust performance in zero-shot settings, notably in Top-10 accuracy, as shown in Table 1.
>
> **Q2:** The method is based on large models with progressive training to establish shape–semantics associations for oracle bone script; this appears task-specific customization. The overall innovation is somewhat limited.
>
> **A2:** Thank you for your valuable feedback. We would like to clarify that our method is not a task-specific customization nor dependent on particular datasets.
>
> First, the proposed progressive radical → pictographic → mutual analysis constitutes a new reasoning curriculum that allows LVLMs to explicitly connect glyph substructures with semantic interpretations. This is not a set of Oracle Bone Script–specific rules, but a generalizable approach that possibly adapted to other ideographic or component-structured writing systems such as the Ancient Chinese script family, Dongba script, Egyptian hieroglyphs.
>
> Second, the Radical–Pictographic Dual Matching introduces a newl analysis-to-matching decoding paradigm that replaces direct generation. This mechanism is model-agnostic, improves robustness in open-set and zero-shot scenarios, and is broadly applicable to recognition tasks where predictions benefit from interpretable intermediate reasoning rather than end-to-end classification.
>
> **Q3:** For undeciphered characters, Top-k outputs provide potential references but still depend on dictionary matching and human experts.
>
> **A3:** We agree that deciphering Oracle Bone Script is an extremely challenging and highly rigorous task, and therefore all existing decipherment methods [2, 3] inevitably require expert validation, regardless of whether they use dictionary matching. This is not a limitation specific to our approach.
>
> For example, the diffusion-based method  OBSD [2] produces reconstructed modern-character images, the generated characters may also be misrecognized by OCR tools. OBSD explicitly describes that the generated modern-character images may contain artifacts or structural inconsistencies and emphasizes that the method “assists” rather than replaces human decipherment. As a result, interpreting the reconstructed glyph still requires human judgment.
>
> Our method does not increase this dependency; instead, it reduces expert workload by providing  radical/pictographic analyses that yields interpretable, semantically grounded Top-k candidates.
>
> Thus, the need for expert verification reflects an intrinsic property of the OBS decipherment task, not a drawback of our method.
>
> ***References:***
>
> [1]  Wang, Pengjie, et al. "Puzzle pieces picker: Deciphering ancient chinese characters with radical reconstruction." *International Conference on Document Analysis and Recognition*. Cham: Springer Nature Switzerland, 2024.
>
> [2]Haisu Guan, et al. "Deciphering Oracle Bone Language with Diffusion Models. " *Proceedings of the 62nd Annual Meeting of* *the Association for Computational Linguistics* *(**ACL**)*, 2024.
>
> [3] Li, Caoshuo, et al. "Oraclefusion: Assisting the decipherment of oracle bone script with structurally constrained semantic typography." *Proceedings of the IEEE/CVF* *International Conference on Computer Vision* *(**ICCV**)*. 2025.

---

> ### Author Response · Authors · 2025-11-25
> **Response to Reviewer 27eb (Part 2)**
>
> **Q4**：Method details need refinement: missing punctuation in formulas, undefined key variables, and unspecified hyperparameter values/settings.
>
> **A4:** Thank you for highlighting these issues. We have carefully reviewed all equations and key variables in the manuscript and made the refinements. For instance, we have added punctuation to Eq. 1 and provided explanations for N and α.
>
> We outlined the specific values and settings in Section 5.1.
>
> **Q5:** Hyperparameter sensitivity analysis.
>
> **A5:** Thank you for your suggestions, which will enhance the completeness of our paper. We have conducted additional sensitivity analysis following the suggestion and the results are provided below.
>
> Sensitivity analysis for the Radical Recognition stage:
>
> We observe that small α makes the triplet constraint insufficiently strong, leading to weaker separation between visually similar radicals, whereas excessively large α introduces optimization instability and degrades performance. In addition, since γ controls the relative contribution of the triplet loss, α and γ should be jointly tuned to ensure balanced optimization and prevent either loss from dominating. The empirical trends in the table support the choice of α=0.25 and γ=5 used in the main experiments.
>
> | α \ γ | 1           | 5           | 10          |
> | ----- | ----------- | ----------- | ----------- |
> | 0.1   | 92.0 / 86.5 | 92.2 / 86.8 | 92.7 / 87.1 |
> | 0.25  | 93.1 / 87.7 | 93.6 / 88.3 | 91.4 / 87.9 |
> | 0.5   | 91.2 / 86.1 | 89.4 / 84.5 | 88.9 / 83.2 |
>
> Sensitivity analysis for the Dual Matching stage:
>
> We have provided ablation studies and matching-parameter analysis (e.g., Top-k and matching variants) in Appendix A.5.
>
> **Q6:** In the radical–pictogram dual matching mechanism, semantic similarity relies on BERT-Score; given BERT pretraining on modern text, how to ensure ancient–modern semantic differences do not bias similarity judgments?
>
> **A6:** Thank you for the thoughtful question. There may be a misunderstanding here. BERT-Score does not suffer from ancient–modern language mismatch in our setting because both the model outputs and all analysis annotations used for matching are written entirely in modern Chinese. In the PD-OBS data pipeline, classical dictionary explanations are normalized and rewritten into concise modern Chinese descriptions before being used for radical and pictographic analysis. As a result, BERT-Score operates within a consistent modern-text semantic space, rather than comparing ancient and modern language.
>
> To provide a more intuitive demonstration of the proposed dataset PD-OBS, we have visualized several data cases in Figure 10 of the revision and more annotation samples are provided in the supplementary material.
>
> **Q7:**  The design of  space patch merger. How to consider the structural features of the oracle bone script and avoid losing detailed and structural information with downsampling.
>
> **A7:** Thank you for the question. The spatial patch merger is designed by stacking the standard patch merger of Qwen2.5-VL multiple times, and consists of RMSNorm layers and MLPs. More details are provided in Appendix A.2.2. Its role is simply to aggregate visual tokens into a compact high-level representation for radical recognition. This is reasonable because radical recognition is essentially a classification task, and visual classification models routinely rely on downsampled high-level semantic features rather than fine-grained details.
>
> For the subsequent pictographic and mutual analysis stages—where detailed structural information is required— we no longer employ this adapter for multiple downsampling operations, ensuring that complex glyph structures of OBS are fully preserved.

---

> > ### Comment · Reviewer_27eb · 2025-11-26
> >
> > Thank you for the detailed response. My main concern with this paper lies in question W2,  whether the proposed method resembles a task-specific customization of large models. The authors have addressed this point with somewhat repetitive but sufficient explanations, which largely alleviates my concern. If the method can be further extended to a broader range of text types in the future, its practical value and applicability would be significantly enhanced. Based on these considerations, I will adjust my score accordingly.

---

### Author Response · Authors · 2025-11-25
**Global Response**

We sincerely thank all reviewers for their constructive feedback. Your comments have greatly helped us improve the clarity, rigor, and presentation of the paper.

Our work introduces a new LVLM-based decipherment pipeline that progressively performs **radical analysis → pictographic analysis → mutual analysis** to bridge the gap between OBS glyphs and modern semantics. The outputs of these stages are then used by our **Radical–Pictographic Dual Matching** mechanism to retrieve decipherment results, enabling **fine-grained interpretability**. The proposed approach achieves competitive recognition and decipherment performance and shows **substantial zero-shot Top-10 accuracy**. While pictographic reasoning has long been discussed in archaeological studies, our contribution lies in transforming this knowledge into an **explicit, trainable, and executable LVLM reasoning framework**.

We also introduce **PD-OBS**, a well-organized benchmark containing OBS images, ancient glyph forms, modern interpretations, and fine-grained radical and pictographic annotations derived from authoritative classical dictionaries. This dataset provides high-quality supervision and a benchmark for interpretable OBS decipherment.

Below we summarize the major clarifications made in our rebuttal:

- **Innovation.** Our novelty lies not in the use of pictographic cues alone, but in integrating radical, pictographic, and mutual analyses into a unified, interpretable LVLM pipeline, and operationalizing OBS decipherment knowledge through structured reasoning and dual matching. This framework offers a new perspective for analyzing and deciphering ideographic or component-structured writing systems.
- **Performance.** Validation accuracy mainly reflects **OBS recognition** (closed-set) capability, while the **zero-shot setting** more accurately reflects decipherment ability, as discussed in Section 5.3.
- **Generalization.** Despite inherent limits, our method demonstrates stronger zero-shot generalization than existing baselines, especially in Top-10 accuracy. Ablation studies further verify the effectiveness of our approach.
- **Expert** **validation***.* In practical archaeological work, expert validation remains indispensable for all existing methods. However, in experimental research settings, zero-shot evaluation on previously deciphered characters provides reliable benchmark data for assessing model capability without incurring substantial expert cost. Therefore, we follow this established practice in our evaluation.
- **Dictionary constraints.** Despite the limitations of dictionary matching, we employ the *Kangxi Dictionary*—which covers all common characters and a substantial number of rare ones—to mitigate these constraints. For OBS characters that lack clear modern equivalents, our method still produces detailed radical and pictographic analyses, offering meaningful reference information for experts.

In response to the reviewers’ suggestions, we made the following revisions:

1. **Additional baselines**, including the PPP composition-based method and the latest commercial LVLMs (GPT-5, Gemini-2.5-Pro) (Table 1 and Table 2).
2. **More precise contribution statements**, removing any absolute or overly strong claims.
3. **Clarification of evaluation settings** (validation vs. zero-shot) in Section 5.3.
4. **New interpretability metrics** (ROUGE-L and METEOR) in Table 2.
5. **Hyperparameter** **sensitivity analysis** in Appendix A.4.
6. **Clarification of radical usage** in Appendix A.2.1.
7. **Detailed evaluation protocol for commercial LVLMs** in Appendix A.2.4.
8. **Analysis of the impact of radical-recognition errors** in Appendix A.5.4.
9. **Expanded limitations** in Appendix A.8.
10. **Visualization of PD-OBS** in Appendix A.9.
11. **Corrections and presentation refinements**, including fixes to Figure 1, Table 1, Equation (1), the EVOBC citation, and the clarity of Figure 4.

All revisions are highlighted in the updated manuscript. We hope our responses resolve the reviewers’ concerns and provide a clearer understanding of our contributions

---

### Meta-Review · Area_Chair_MPA1 · 2026-01-04

**Summary:**

Multiple reviewers raised the concern of limited novelty and contribution of the proposed method [R27eb, Ru3mu, RDqfC, RxkzS]. The method lacks state-of-the-art performance [Ru3mu, RDqfC, RxkzS] and should be compared to more recent vision-language models [LVLMs] [Ru3mu, RxkzS]. Reviewers also pointed out the paper's overstatement of being the "first" [RDqfC], and raised concerns about the limitations, e.g., its narrow scope compared to baselines [RDqfC, RxkzS]. Additionally, reviewers asked for sensitivity analysis regarding the hyperparameters, and there are several issues with the method's details, formulas, and figures [R27eb, RDqfC, RxkzS].

**Reviewer Concerns:**

During the rebuttal, the concerns regarding comparisons with modern models are addressed with additional results provided. The method’s details and clarification were added. However, the AC still finds that there are more changes necessary to the writing, e.g., L_{ce} in Eq. 2 is not defined. Certain statements are not fully accurate, e.g., “We design a triplet loss”; it is not immediately clear to the reader which part is “designed” by the authors; it looks like a standard triplet loss. From the AC’s perspective, concerns regarding novelty and the narrow scope remain outstanding.

**Reviewer Scores:**

While the authors summarized that multiple reviewers have changed their score to 6. The AC has no way to validate such a claim. The AC was only able to verify that Reviewer DqfC “ I decide to raise my rating” (I.e., > 4) and that Reviewer 27eb “I will adjust my score accordingly.” (i.e., >4). Given that part of the concerns were not fully addressed, the AC believes that the reviewer could have raised their score, but it does not seem that they will be strongly supporting an acceptance. In particular, the novelty and contribution of this work are quite limited from the perspective of ICLR, specifically in terms of the introduced techniques and relevance to the conference.

---

### Decision · Program_Chairs · 2026-01-26

Reject